# IRIS: INTRINSIC REWARD IMAGE SYNTHESIS

## ABSTRACT

Despite the success of Reinforcement Learning from Human Feedback (RLHF) in language reasoning, its application to autoregressive Text-to-Image (T2I) generation is often constrained by the limited availability of human preference data. This paper explores how an autoregressive T2I model can learn from internal signals without relying on external rewards or labeled data. Contrary to recent findings in text generation, we show that maximizing self-uncertainty, rather than self-certainty, improves image generation. We observe that this is because autoregressive T2I models with low uncertainty tend to generate simple and uniform images, which are less aligned with human preferences. Based on these observations, we propose **IRIS** (**I**ntrinsic **R**eward **I**mage **S**ynthesis), the first framework to improve autoregressive T2I models with reinforcement learning using only an intrinsic reward. Empirical results demonstrate that applying IRIS to autoregressive T2I models achieves performance that is competitive with or superior to external rewards.

## 1 INTRODUCTION

Reinforcement Learning (RL) has proven to be highly effective in advancing the reasoning capabilities of large language models, particularly in verifiable domains such as mathematics and programming (Shao et al., 2024; Hurst et al., 2024; Guo et al., 2025a; Hu et al., 2025). Motivated by these advances, a similar RL-based alignment approach is now being explored for text-to-image models (Betker et al., 2023; Team et al., 2023; Esser et al., 2024a). However, applying RL here is more challenging due to the lack of verifiable rule-based rewards—unlike math or code, the quality of a visual output is inherently subjective and hard to evaluate automatically. Existing methods in this area either build an image reward model from human preferences or use automated rewards from specialized models, such as object detectors (Yan et al., 2024) or Visual Question Answering (VQA) systems (Jiang et al., 2025). However, the former approach is bottlenecked by the scalability and subjectivity of human labeling, while the latter is often domain-specific and struggles to generalize beyond the narrow settings for which it was trained.

Recently, several works have shown that text-based applications, including mathematical and code reasoning, can be improved by maximizing self-certainty via RL (Zhao et al., 2025b; Zhang et al., 2025a). This concept is particularly appealing for text-to-image generation, given the inherent difficulty of defining an explicit reward model. Motivated by this, we aim to answer the following question:

> *Is it possible to design a more general method for text-to-image generation using only intrinsic signals—without relying on human-labeled data or domain-specific heuristics?*

In this work, we use the multimodal LLM, Janus-Pro (Chen et al., 2025), for the text-to-image generation task due to their strong instruction-following capabilities. However, contrary to previous works in the text generation (Zhao et al., 2025b; Zhang et al., 2025a), our research reveals that maximizing self-certainty is detrimental to text-to-image synthesis. As illustrated in the "self-certainty" column of Fig. 1, maximizing a model's self-certainty by reinforcement learning impairs its image generation. Furthermore, we discover that more self-confident models tend to generate uniform and simplistic images, whereas less self-confident models produce more visually rich and colorful images that are better aligned with human preferences. We thus claim that:

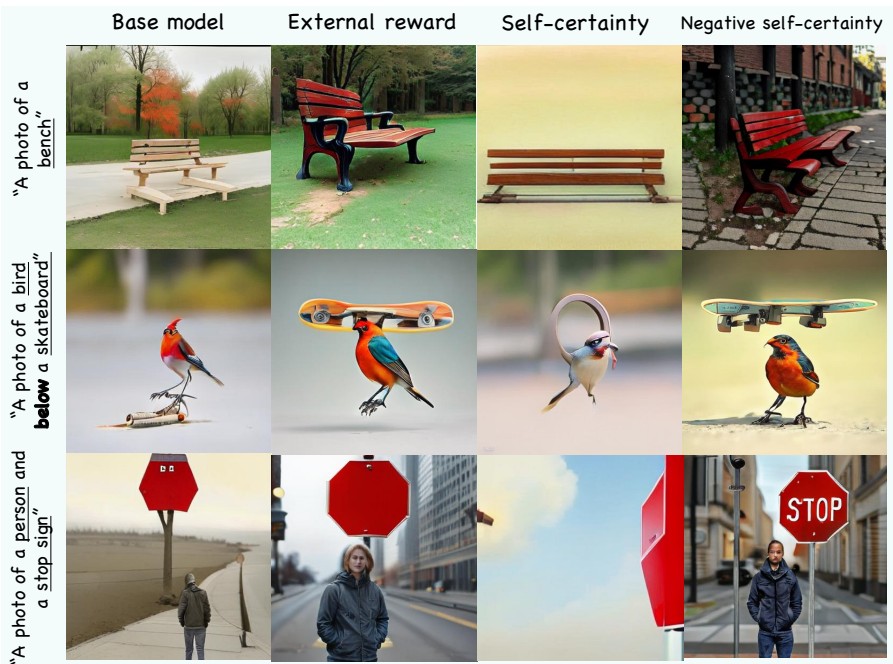

Figure 1: We perform reinforcement learning (RL) fine-tuning on Janus-Pro using three reward schemes: (1) external reward (pretrained image reward models, etc), (2) self-certainty reward, and (3) negative self-certainty reward. The self-certainty is computed as the negative cross-entropy between the model's output distribution and a uniform distribution—where higher self-certainty indicates greater model self-certainty. The figure presents results across three tasks: (i) single-object generation, (ii) spatial generation, and (iii) two-object generation. We observe that increased self-certainty typically results in more uniform and less visually diverse images, while lower self-certainty tends to generate images with richer visual features that are more preferred by humans. Please refer to Appx. B.6 for more visualized results.

*Maximizing self-certainty improves language reasoning,*
*while minimizing self-certainty improves text-to-image generation.*

To verify the hypothesis quantitatively, we compute the self-certainty measured by the KL divergence between the uniform distribution and the model's output distribution. We train an LLM and a multimodal LLM, i.e., Qwen2.5-1.5B-Instruct and Janus-Pro-1B respectively, on verifiable external rewards by GRPO (Shao et al., 2024) and monitor the self-certainty on the text and image tokens respectively. Results in Fig 2 shows that RL alignment with external reward models continuously increases the self-certainty of the LLM in math reasoning tasks, but decreases the self-certainty of the multimodal LLM in the text-to-image generation. This indicates that less self-confident multimodal LLMs will generate images with higher rewards. We give a detailed description of the experiment settings in Appx. B.1.

Based on this observation, we introduce **IRIS** (**I**ntrinsic **R**eward **I**mage **S**ynthesis), a reinforcement learning framework that leverages

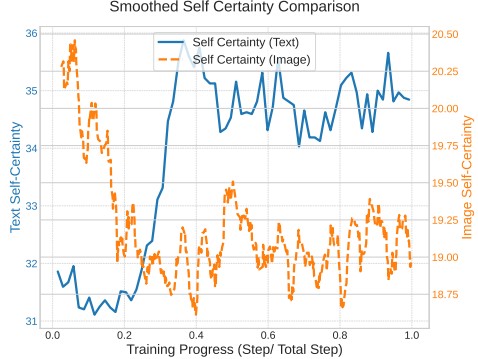

Figure 2: **Self-Certainty** on image tokens in the Janus-Pro-1B (orange line, right $y$-axis), and on text tokens in the Qwen2.5-1.5B-Instruct (blue line, left $y$-axis). RL alignment with external reward models continuously increases the self-certainty of the LLM in the math reasoning, but decreases the self-certainty of the multimodal LLM in the text-to-image generation.

the model's the Negative Self-Certainty (NSC) as the reward signal. We note that IRIS doesn't need any human knowledge or external verifiers, and is agnostic to the model architecture or dataset. Remarkably, we find that our IRIS itself can significantly *enhance the reasoning capabilities of T2I models*. We adopt the CoT reasoning to generate detailed image description prior to subsequent image synthesis, termed as semantic CoTs (Jiang et al., 2025). On Janus-Pro 1B models, IRIS boosts the performance by 9.1%, 13.3%, and 28.8% on GenEval (Ghosh et al., 2023), T2I-CompBench (Huang et al., 2025) and WISE (Niu et al., 2025) respectively, showing that IRIS improves reasoning and planning capabilities in the T2I models. See Sec. 4.2 for more details.

Our contributions can be summarized in the following:

- We propose IRIS, the first RL-based alignment method for text-to-image generation using **only an intrinsic reward**. We minimize the self-certainty of the T2I models.
- We observe and confirm that the model's self-certainty exhibits task-dependent behaviors. Specifically, higher self-certainty benefits the model in domains requiring objective reasoning (e.g., mathematical and code reasoning), but lower self-certainty benefits the model in subjective generation tasks (e.g., text-to-image generation).
- Empirically, we show that IRIS can improve the autoregressive T2I models without any external supervision. Post-training by IRIS achieves competitive performance compared to training with external rewards. We call for better design of training algorithm and external reward models that are beyond the intrinsic capabilities of the base T2I model.

## 2 RELATED WORK

**Reinforcement learning in image generation models** Reinforcement learning plays a key role in enhancing the performance of modern text-to-image generation models (Betker et al., 2023; Team et al., 2023; Wallace et al., 2024; Esser et al., 2024a). Early efforts primarily focused on training image reward models using human-labeled preference data (Xu et al., 2023; Wu et al., 2023; Xu et al., 2024). To reduce reliance on manual annotation, subsequent work has explored automated reward generation. For example, Yan et al. (2024) leverage existing automatic scoring methods such as the Grounding DINO detector (Liu et al., 2024). Guo et al. (2025b) fine-tunes LLaVa-OneVision (Li et al., 2024b) to evaluate the alignment between the prompt and the generated image. Similarly, Jiang et al. (2025) utilize Visual Question Answering (VQA) models to provide feedback signals during training. However, our method get rid of the external models and guide the generation model with its intrinsic signals, making it adaptable to many scenarios.

**Reinforcement learning with intrinsic reward** Recent work in LLMs explores RLIF as a means to reduce reliance on human preference data or domain-specific verifiers. For instance, Zhang et al. (2025a) and Agarwal et al. (2025) propose minimizing entropy as a form of reasoning incentivization. Building on this idea, Zhao et al. (2025b) introduce a self-certainty signal—defined as the cross-entropy between the output token distribution and a uniform distribution to guide RL training, reporting improved performance. In related efforts, Prasad et al. (2024) and Zuo et al. (2025) generate multiple rollouts and leverage majority-vote outcomes to estimate advantages. To the best of our knowledge, we are the first to successfully train text-to-image generation models without external reward supervision. Our key insight lies in the observation that visual generative models exhibit lower self-certainty when producing visually rich and semantically meaningful images. This contrasts with findings in the LLM domain, where higher model self-certainty has been associated with better performance (Zhang et al., 2025a; Agarwal et al., 2025; Zhao et al., 2025b).

## 3 METHOD

### 3.1 RL FINETUNING OF LLMS

In RL finetuning of LLMs, the LLM policy $\pi_\theta$ is optimized to maximize some reward function $r$. Given the input query $q$, the generated output $o$, the reference policy $\pi_{\text{ref}}$, and the KL regularization coefficient $\beta$, the objective to optimize is

$$\max_{\pi_\theta} \mathbb{E}_{o \sim \pi_\theta(\cdot|q)} \left[ r(o|q) - \beta \text{KL}(\pi_\theta(o|q)) \| \pi_{\text{ref}}(o|q)) \right] . \tag{1}$$

In this paper, we denote by $o_t$ the $t$-th token of the output $o$, and $o_{<t}$ the first $t-1$ tokens of the output $o$. We use $\text{KL}(p\|q)$ to denote the KL diveregnce of the distribution $p$ and $q$.

Currently, RL finetuning mainly consists of reinforcement learning from human feedback (RLHF, Christiano et al. (2017); Ouyang et al. (2022); Kaufmann et al. (2023)), reinforcement learning from verifiable reward (RLVR, Lambert et al. (2023)), and reinforcement learning from intrinsic reward (RLIF, Zhang et al. (2025a); Agarwal et al. (2025); Zhao et al. (2025b)).

- **RLHF**. The reward function is typically learned explicitly or implicitly from human's preferences.

- **RLVR**. The reward function is verifiable. For example, in mathematical problem solving, the reward is 1 if the output solution is correct and 0 otherwise. In text-to-image generation, the reward model can be the object detector (Liu et al., 2024).

- **RLIF**. The reward function is an intrinsic signal derived from the model's intrinsic state.

However, existing RL finetuning methods mainly focus on text outputs, either in standard LLMs or iamge-to-text language models. In this paper, we focus on RL finetuning text-to-image generation with intrinsic feedback.

### 3.2 IRIS AND REWARD DESIGN

During training, given the prompt $q$, we first generate a semantic-level text description and then use it to guide the visual generation. We denote the output by $o$, which contains both text and image tokens. Conditioned on the prompt $q$ and the output before the $t$-th position $o_{<t}$, we define Self-Certainty (SC, Zhao et al. (2025b)) and Negative Self-Certainty (NSC) at position $t$ by

$$\text{SC}(o_t|q, o_{<t}) := +\text{KL}(U\|\pi_\theta(o_t|q, o_{<t})), \quad \text{and} \quad \text{NSC}(o_t|q, o_{<t}) := -\text{SC}(o_t|q, o_{<t}), \quad (2)$$

where $U$ denotes the uniform distribution on the vocabulary.

The output token $o_t$ could be a text or image token in the multimodal LLMs. A natural question is whether to use SC or NSC as the intrinsic reward on the text or image tokens. If we use SC as the intrinsic reward in the RL objective, maximizing the reward means improving the self-certainty, and if we use NSC, maximizing the reward means improving the self-uncertainty. For the image tokens, as we discussed in Fig. 1, overly confident models usually generate uniform and plain figures, whereas models with a moderate confidence can generate images with richer and more diverse features. For text tokens, we argue that maximizing NSC encourages the generation of more diverse semantic CoTs, thereby facilitating better exploration during training. To verify our claim quantitatively, we carefully design the ablation study in Sec. 4.3, which shows that using the NSC as the intrinsic reward in both text and image tokens can achieve the best results. This is in clear contrast to previous works on pure text generation models, which maximize SC (Zhao et al., 2025b; Zhang et al., 2025a). In conclusion, we use NSC as our intrinsic reward in IRIS.

We define self-certainty by the forward KL divergence, which encourages mode-covering behavior by rewarding probability distributions that cover multiple plausible outcomes. This stands in contrast to metrics like entropy (backward KL), which are mode-seeking and favor a single high-probability output. Specifically, self-certainty mitigates the common bias against longer sequences found in perplexity and entropy-based measures, making it a more robust metric for a model's intrinsic self-certainty (Fang et al., 2024; Kang et al., 2025). Its practical value is supported by the recent work that it can serve as a powerful intrinsic reward to guide language model's learning across different domains (Zhao et al., 2025b). We also show in the ablation study of Sec. 4.3, that forward KL is better than backward KL in IRIS self-uncertainty computation.

We optimize the objective function from Eq. (1) by applying Group-wise Relative Policy Optimization (GRPO) to the IRIS reward. GRPO's optimization process relies on sampling multiple candidates to inform policy updates. Specifically, for each query $q \sim P(Q)$, we generate a set of $G$ outputs $\{o_1, \ldots, o_G\}$ using a fixed behavior policy $\pi_{\theta_{\text{old}}}$. The relative rewards of these outputs are then used to estimate advantages, guiding the update for the target policy $\pi_\theta$ by maximizing the

following objective:

$$\mathcal{J}_{\text{GRPO}}(\theta) = \mathbb{E}_{q \sim P(Q), O = \{o_i\}_{i=1}^{G} \sim \pi_{\theta_{\text{old}}}(\cdot|q)}$$

$$\frac{1}{G} \sum_{i=1}^{G} \frac{1}{|o_i|} \sum_{t=1}^{|o_i|} \left\{ \min \left( c_{i,t}(\theta) \hat{A}_{i,t}, \ \text{clip}\left( c_{i,t}(\theta), 1 - \epsilon, 1 + \epsilon \right) \hat{A}_{i,t} \right) - \beta \text{KL}\left( \pi_\theta \| \pi_{\text{ref}} \right) \right\},$$

where ratios $c_{i,t}$ are defined by $c_{i,t}(\theta) = \frac{\pi_\theta(o_{i,t}|q,o_{i,<t})}{\pi_{\theta_{\text{old}}}(o_{i,t}|q,o_{i,<t})}$, and the advantage can be estimated by

$$u_i = \sum_t \text{NSC}(o_{i,t}|q, o_{i,<t}), \quad \hat{A}_{i,t} = \frac{u_i - \text{mean}(\{u_1, u_2, \cdots, u_G\})}{\text{std}(\{u_1, u_2, \cdots, u_G\})}.$$

In the autoregressive T2I models, the generated output $o_i$ consists of text and image tokens, denoted by Let $o_{i,\text{text}}$ and $o_{i,\text{img}}$ respectively. We simply concatenate $o_{i,\text{text}}$ and $o_{i,\text{img}}$ to be $o_t$. Therefore, our objective is to maximize the uncertainty (NSC) for both text and image tokens. We validate this choice in the ablation study (Sec. 4.3, Fig. 6 and 7).

Notably, maximizing uncertainty for text tokens appears to contradict our observation in Fig. 2, where we found that training on reasoning tasks decreases uncertainty. We speculate this discrepancy arises because math reasoning requires precise thought generation. In contrast, our T2I setting first generates descriptive and explorative text from a given prompt before synthesizing the image. This need for textual exploration is supported by recent work on information-seeking agents Team et al. (2025), which similarly observed that as the agent's reward increases, its output entropy (and thus uncertainty) also increases.

## 4 EXPERIMENTS

### 4.1 EXPERIMENT CONFIGURATION

To evaluate the effectiveness of IRIS, we primarily follow the protocol in T2I-R1 Jiang et al. (2025).Our experiments focus on fine-tuning Janus-Pro (Chen et al., 2025), using Generative Relative Process Optimization (GRPO) (Guo et al., 2025a). Key hyperparameters include: a learning rate of $1 \times 10^{-6}$, a maximum prompt length of 512 tokens, and a maximum completion length of 1024 tokens. We use an effective batch size of 8, achieved with a per-device batch size of 1, a dataparallel width of 4 GPUs, and 2 gradient accumulation steps. The GRPO algorithm is configured with a KL divergence coefficient ($\beta$) of 0.01. For our text-to-image tasks, Janus-Pro models first generate semantic Chains of Thought (CoTs) before creating the final image.

To comprehensively assess our model's capabilities, we evaluate it against three diverse benchmarks, each designed to test different aspects of text-to-image generation. First, GenEval (Ghosh et al., 2023) provides an object-centric evaluation, focusing on fundamental abilities such as correctly rendering single or multiple objects, their colors, counts, and positions. Second, T2I-CompBench (Huang et al., 2023) targets compositional understanding, specifically assessing the model's capacity for attribute binding (e.g., color, shape, texture) and its handling of both spatial and non-spatial relationships between objects. Finally, WISE (World Knowledge-Informed Semantic Evaluation, Niu et al. (2025)) measures the model's ability to apply real-world knowledge, evaluating performance on prompts requiring cultural common sense, spatio-temporal reasoning, and an understanding of natural sciences. Collectively, these benchmarks provide a multi-faceted evaluation of our model's performance, ranging from basic object composition to complex, knowledge-based semantic interpretation. We give a more detailed description of the benchmarks in Appx. A.1. Following previous benchmark results, we round the scores to two decimal places in the GenEval and WISE benchmarks, and four decimal places in the T2I-CompBench benchmark.

We will use the four external reward models to train the multimodal LLM as the baseline (T2I-R1, (Jiang et al., 2025)). Trained from human aesthetic preferences, the Human Preference Model (HPSv2, Wu et al. (2023)) assesses the overall aesthetic appeal and visual quality from the human perspective. To evaluate compositional accuracy, we use the GroundingDINO (DINO, Liu et al. (2024)) object detector to verify the existence, count, and spatial arrangement of specified objects. Complementing this, a Visual Question Answering model GIT (Wang et al., 2022) question the

image to confirm specific attributes, such as color and texture. Finally, an Output Reward Model (ORM, Guo et al. (2025b)), a fine-tuned Large Multimodal Model, provides a holistic judgment on the alignment between the prompt and the generated image. We give a more detailed description of the external rewards in Appx. A.2.

We identify a key inconsistency in the official implementation of T2I-R1 (Jiang et al., 2025). Janus and Janus-Pro models use different chat templates: Janus models use keys "User" and "Assistant", but Janus-Pro models use keys "<|User|>" and "<|Assistant|>". Jiang et al. (2025) uses Janus model's chat template to train and evaluate the Janus-Pro models. In this paper, we will use the correct chat template to train and evaluate the Janus-Pro models, so numerical results reported in our paper are different from those in Jiang et al. (2025).

## 4.2    MAIN RESULTS

We evaluate our results on three series of models: (1) Janus-Pro base models, (2) Janus-Pro trained by external rewards. (3) Janus-Pro trained by IRIS, where we generate 8 text strings per query and subsequent 1 image per text string in GRPO's advantage computation. Fig. 3 shows the main results on GenEval, T2I-CompBench and WISE on the first 800 steps of training. Tab. 1 reports the best result of different methods among the checkpoints from 100 step to 800 step on the three benchmarks. We found that IRIS boosts the performance of the Janus-Pro 1B model by 9.1%, 13.3%, and 28.8% on GenEval, T2I-CompBench and WISE respectively, and Janus-Pro 7B model by 1.3%, 1.8%, and 6.5%, achieving results comparable to its counterpart that uses an external reward. The larger performance gain on WISE benchmark can be attributed by the novelty and difficulty of this benchmark, and relative smaller performance gain of RL finetuning on 7B models can be attributed to the stronger capability of larger base models. Importantly, our method does not rely on any external knowledge or domain specific verifier and can be easily adapted to any scenario, highlighting its broad applicability and potential. Please refer to Appx. B.3 for detailed subscores in the three benchmarks.

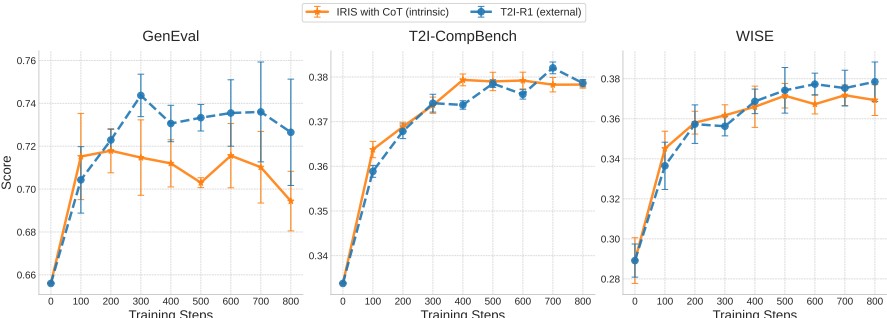

Figure 3: **Main results** of Janus-Pro 1B on GenEval, T2I-CompBench and WISE. On T2I-CompBench and WISE, IRIS can achieve comparable results with T2I-R1.

**Emergence of long-form reasoning**    Jiang et al. (2025) shows that using external rewards can guide the model to generate meaningful semantic CoTs. We show that that using intrinsic rewards can also guide the model to generate meaningful semantic CoTs that helps the image generation. In Fig. 4, we present an example of our semantic CoTs in training, which provides concise and meaningful details that could be useful in training to enhance the image diversity. More CoT examples could be found in Fig. 13 in Appx. B.4.

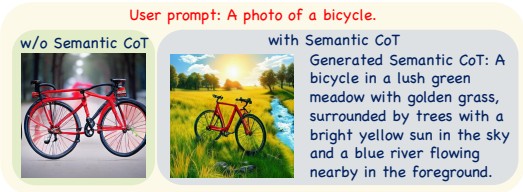

Figure 4: **Visualization of semantic CoTs**. The left one is training without semantic CoTs, and the right one is training with semantic CoTs. Training with semantic CoTs improve image generation.

**Intrinsic rewards incentive general T2I abilities**    The NSC reward in IRIS provides an intrinsic, token-level signal for guided image generation. IRIS surpasses the T2I-R1 on 1B models in

Table 1: "Und." and "Gen." denote "understanding" and "generation", respectively. We report the scores of the best checkpoint (measured by the average performance) of the T2I-R1 and IRIS.

(a) **GenEval.**

| Type | Method | Single Obj.↑ | Two Obj.↑ | Counting↑ | Colors↑ | Position↑ | Color Attri.↑ | Overall↑ |
|---|---|---|---|---|---|---|---|---|
| Gen. Only | SD-v1 (Rombach et al., 2022) | 0.97 | 0.38 | 0.35 | 0.76 | 0.04 | 0.06 | 0.43 |
| | SD-v2 (Rombach et al., 2022) | 0.98 | 0.51 | 0.44 | 0.85 | 0.07 | 0.17 | 0.50 |
| | PixArt-$\alpha$ (Chen et al., 2023b) | 0.98 | 0.50 | 0.44 | 0.80 | 0.08 | 0.07 | 0.48 |
| | SDXL (Podell et al., 2023) | 0.98 | 0.74 | 0.39 | 0.85 | 0.15 | 0.23 | 0.55 |
| | FLUX (Labs, 2024; Yang et al., 2024b) | 0.98 | 0.81 | 0.74 | 0.79 | 0.22 | 0.45 | 0.64 |
| | SD3-Medium (Esser et al., 2024b) | 0.99 | 0.94 | 0.72 | 0.89 | 0.33 | 0.60 | 0.74 |
| Und. & Gen. | Show-o (Xie et al., 2024) | 0.98 | 0.80 | 0.66 | 0.84 | 0.31 | 0.50 | 0.68 |
| | SEED-X (Ge et al., 2024) | 0.97 | 0.58 | 0.26 | 0.80 | 0.19 | 0.14 | 0.49 |
| | Janus-Pro-1B (Chen et al., 2025) | 0.94 | 0.71 | 0.42 | 0.82 | 0.52 | 0.51 | 0.66 |
| | Janus-Pro-1B + T2I-R1 (Jiang et al., 2025) | $0.99_{\pm0.01}$ | $0.85_{\pm0.03}$ | $0.50_{\pm0.03}$ | $0.86_{\pm0.01}$ | $0.64_{\pm0.05}$ | $0.63_{\pm0.02}$ | $0.75_{\pm0.01}$ |
| | Janus-Pro-1B + IRIS (Ours) | $0.99_{\pm0.01}$ | $0.85_{\pm0.01}$ | $0.41_{\pm0.03}$ | $0.88_{\pm0.02}$ | $0.66_{\pm0.04}$ | $0.51_{\pm0.03}$ | $0.72_{\pm0.01}$ |
| | Janus-Pro-7B (Chen et al., 2025) | 0.98 | 0.89 | 0.49 | 0.89 | 0.69 | 0.62 | 0.76 |
| | Janus-Pro-7B + T2I-R1 (Jiang et al., 2025) | $1.00_{\pm0.01}$ | $0.91_{\pm0.03}$ | $0.55_{\pm0.03}$ | $0.91_{\pm0.01}$ | $0.69_{\pm0.04}$ | $0.62_{\pm0.03}$ | $0.78_{\pm0.01}$ |
| | Janus-Pro-7B + IRIS (Ours) | $0.99_{\pm0.01}$ | $0.91_{\pm0.03}$ | $0.52_{\pm0.06}$ | $0.88_{\pm0.03}$ | $0.73_{\pm0.05}$ | $0.61_{\pm0.03}$ | $0.77_{\pm0.03}$ |

(b) **T2I-CompBench**

| Type | Method | Attribute Binding | | | Object Relationship | | Complex↑ |
|---|---|---|---|---|---|---|---|
| | | Color ↑ | Shape↑ | Texture↑ | 2D-Spatial↑ | Non-Spatial↑ | |
| Gen. Only | SD-v1 (Rombach et al., 2022) | 0.3765 | 0.3576 | 0.4156 | 0.1246 | 0.3079 | 0.3080 |
| | SD-v2 (Rombach et al., 2022) | 0.5065 | 0.4221 | 0.4922 | 0.1342 | 0.3127 | 0.3386 |
| | PixArt-$\alpha$ (Chen et al., 2023a) | 0.6690 | 0.4927 | 0.6477 | 0.2064 | 0.3197 | 0.3433 |
| | SDXL (Podell et al., 2023) | 0.5879 | 0.4687 | 0.5299 | 0.2133 | 0.3119 | 0.3237 |
| | FLUX.1 (Labs, 2024) | 0.7407 | 0.5718 | 0.6922 | 0.2863 | 0.3127 | 0.3703 |
| | SD3-Medium (Esser et al., 2024b) | 0.8132 | 0.5885 | 0.7334 | 0.3200 | 0.3140 | 0.3771 |
| Und. & Gen. | Show-o (Xie et al., 2024) | 0.56 | 0.41 | 0.46 | 0.20 | 0.30 | 0.29 |
| | Show-o + PARM (Guo et al., 2025b) | 0.75 | 0.56 | 0.66 | 0.29 | 0.31 | 0.37 |
| | Janus-Pro-1B (Chen et al., 2025) | 0.4922 | 0.2752 | 0.3965 | 0.1284 | 0.2964 | 0.3338 |
| | Janus-Pro-1B + T2I-R1 (Jiang et al., 2025) | $0.7924_{\pm0.0018}$ | $0.4822_{\pm0.0088}$ | $0.6691_{\pm0.0038}$ | $0.3153_{\pm0.0035}$ | $0.3064_{\pm0.0003}$ | $0.3820_{\pm0.0009}$ |
| | Janus-Pro-1B + IRIS (Ours) | $0.7946_{\pm0.0002}$ | $0.4788_{\pm0.0082}$ | $0.6756_{\pm0.0035}$ | $0.2909_{\pm0.0042}$ | $0.3101_{\pm0.0002}$ | $0.3793_{\pm0.0013}$ |
| | Janus-Pro-7B (Chen et al., 2025) | 0.6518 | 0.4364 | 0.5529 | 0.1948 | 0.3097 | 0.3845 |
| | Janus-Pro-7B + T2I-R1 (Jiang et al., 2025) | $0.8015_{\pm0.0002}$ | $0.5661_{\pm0.0070}$ | $0.7081_{\pm0.0051}$ | $0.3246_{\pm0070}$ | $0.3090_{\pm0.0002}$ | $0.3992_{\pm0.0019}$ |
| | Janus-Pro-7B + IRIS (Ours) | $0.7921_{\pm0.0014}$ | $0.5155_{\pm0.0083}$ | $0.6608_{\pm0.0025}$ | $0.2875_{\pm0.0061}$ | $0.3100_{\pm0.0003}$ | $0.3916_{\pm0.0024}$ |

(c) **WISE**

| Type | Method | Cultural↑ | Spatio-Temporal | | Natural Science | | | Overall↑ |
|---|---|---|---|---|---|---|---|---|
| | | | Time↑ | Space↑ | Biology↑ | Physics↑ | Chemistry↑ | |
| Gen. Only | SD-v1 (Rombach et al., 2022) | 0.34 | 0.35 | 0.32 | 0.28 | 0.29 | 0.21 | 0.32 |
| | SD-v2 (Rombach et al., 2022) | 0.30 | 0.38 | 0.35 | 0.33 | 0.34 | 0.21 | 0.32 |
| | PixArt-$\alpha$ (Chen et al., 2023a) | 0.45 | 0.50 | 0.48 | 0.49 | 0.56 | 0.34 | 0.47 |
| | SD-XL (Podell et al., 2023) | 0.43 | 0.48 | 0.47 | 0.44 | 0.45 | 0.27 | 0.43 |
| | FLUX.1 (Labs, 2024) | 0.48 | 0.58 | 0.62 | 0.42 | 0.51 | 0.35 | 0.50 |
| Und. & Gen. | Orthus-7B (Kou et al., 2024) | 0.23 | 0.31 | 0.38 | 0.28 | 0.31 | 0.20 | 0.27 |
| | Show-o (Xie et al., 2024) | 0.28 | 0.36 | 0.40 | 0.23 | 0.33 | 0.22 | 0.30 |
| | VILA-U (Wu et al., 2024) | 0.26 | 0.33 | 0.37 | 0.35 | 0.39 | 0.23 | 0.31 |
| | Janus-Pro-1B (Chen et al., 2025) | 0.24 | 0.28 | 0.43 | 0.28 | 0.35 | 0.15 | 0.28 |
| | Janus-Pro-1B + T2I-R1 (Jiang et al., 2025) | $0.35_{\pm0.01}$ | $0.42_{\pm0.02}$ | $0.49_{\pm0.02}$ | $0.36_{\pm0.02}$ | $0.43_{\pm0.02}$ | $0.22_{\pm0.01}$ | $0.38_{\pm0.01}$ |
| | Janus-Pro-1B + IRIS (Ours) | $0.33_{\pm0.02}$ | $0.39_{\pm0.01}$ | $0.49_{\pm0.01}$ | $0.36_{\pm0.01}$ | $0.45_{\pm0.01}$ | $0.22_{\pm0.01}$ | $0.37_{\pm0.01}$ |
| | Janus-Pro-7B (Chen et al., 2025) | 0.44 | 0.49 | 0.60 | 0.45 | 0.52 | 0.27 | 0.46 |
| | Janus-Pro-7B + T2I-R1 (Jiang et al., 2025) | $0.48_{\pm0.02}$ | $0.52_{\pm0.01}$ | $0.61_{\pm0.02}$ | $0.48_{\pm0.03}$ | $0.56_{\pm0.01}$ | $0.29_{\pm0.01}$ | $0.50_{\pm0.01}$ |
| | Janus-Pro-7B + IRIS (Ours) | $0.48_{\pm0.01}$ | $0.51_{\pm0.01}$ | $0.59_{\pm0.02}$ | $0.46_{\pm0.01}$ | $0.55_{\pm0.01}$ | $0.28_{\pm0.01}$ | $0.48_{\pm0.01}$ |

categories *biology*, *physics*, *chemistry* within *natural science* of the WISE benchmark, whereas T2I-R1 demonstrates advantages in tasks related to aesthetics and spatial relations, such as *counting* and *color attribution* in GenEval, *shape*, *texture*, and *2D-spatial* in T2I-CompBench, and *spatio-temporal* in WISE. We attribute this to the fact that T2I-R1 training uses human aesthetic preferences (HPSv2) and spatial relations (DINO) in the external rewards. Besides, VQA-based rewards, i.e., GIT and ORM, can help align the images with the prompts in these tasks. The inclusion of these external rewards can thus benefit related downstream tasks. However, in downstream tasks irrelevant to the domain of external rewards, like *natural science*, T2I-R1 will lose its advantage. In our external rewards, HPSv2 and DINO are irrelevant to the natural science prompts, and reward models in VQA-based GIT and ORM also lack sufficient information to score the generated images in these categories. Therefore, we argue that incentivizing a model's inherent ability with intrinsic rewards leads to effective and general exploration and learning than relying on specific external signals.

### 4.3 ABLATION STUDY

**Evaluation metrics** We use the four external rewards, namely HPSv2 (Wu et al., 2023), DINO (Liu et al., 2024), ORM (Guo et al., 2025b), and GiT (Wang et al., 2022) introduced in Sec. 4.1, to evaluate the image generation in the ablation studies. Previously, we used these reward models to train the baseline T2I-R1 model. However, in our ablation studies on IRIS, we never use these reward models in the training objectives, so they can be simple and unbiased metrics to evaluate the performance. We regenerate 553 GenEval prompts to synthesis the images. For each prompt, we generate four images to reduce noise. We report four averaged rewards on these images in the ablation studies.

**Training with or without semantic CoTs** T2I-R1 (Jiang et al., 2025) suggests training with semantic CoTs benefits training by external rewards. We show that training with semantic CoTs also benefits training by intrinsic rewards. We consider two series of models: (1) Janus-Pro trained by IRIS, but without semantic CoTs, where we generate 8 images per query in GRPO's advantage computation. (2) Janus-Pro trained by IRIS (with semantic CoTs), where we generate 8 text strings per query and subsequent 1 image per text string in GRPO's advantage computation. Results in 5 show that training IRIS with semantic CoTs consistently outperforms being without semantic CoTs. In Fig. 14, we present some generated figures of training with and without CoTs.

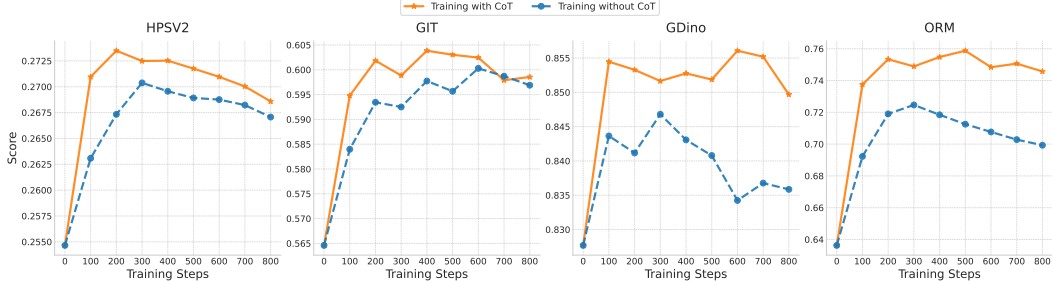

Figure 5: **Ablation study**: training with CoT outperforms training without CoT

**Maximize or minimize image self-certainty** To determine whether image self-certainty should be maximized or minimized, we conduct three experiments: (1) minimizing text self-certainty only, (2) minimizing both text and image self-certainty (IRIS), and (3) minimizing text self-certainty and maximizing image self-certainty. We run GRPO for 800 steps and evaluate four external rewards every 100 steps. In Figure 6, we show that minimizing both text and image self-certainty improves performance, whereas minimizing text self-certainty alone has little effect. Interestingly, maximizing image self-certainty actually degrades performance, causing a rapid drop, which supports our claim: lower self-certainty improves image generation.

**Maximize or minimize text self-certainty** To evaluate whether text self-certainty should be maximized or minimized, we conduct three experiments: (1) minimizing image self-certainty only, (2)

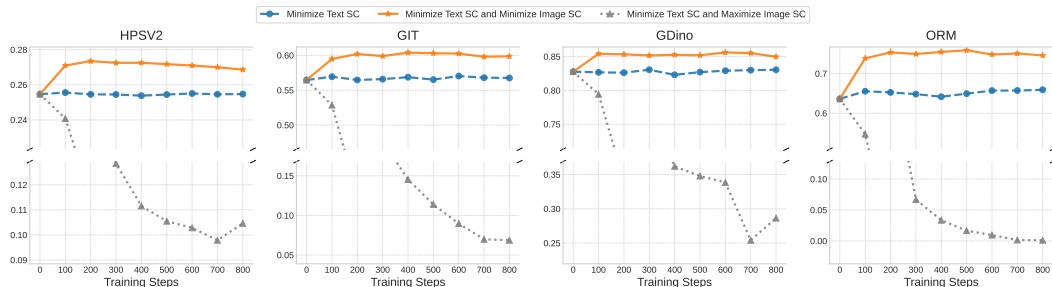

Figure 6: **Ablation study**: minimizing image self-certainty outperforms maximizing it.

minimizing both text and image self-certainty (IRIS), and (3) minimizing image self-certainty and maximizing text self-certainty. In Figure 7, we show that minimizing image self-certainty only achieves comparable performance in the early stages, however, it deteriorates rapidly after 200 steps. Meanwhile, minimizing text self-certainty always outperforms maximizing text self-certainty. This verifies our claim that maximizing text self-certainty discourages the model from exploring diverse semantic CoTs, thereby impairing its reasoning ability. In conclusion, minimizing text self-certainty proves to be a better strategy.

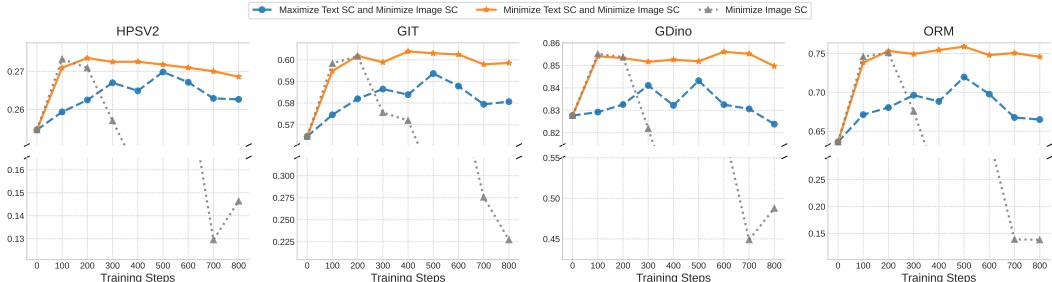

Figure 7: **Ablation study**: minimizing text self-certainty outperforms maximizing it.

**Forward or backward KL**    We consider the backward KL divergence formulation of IRIS reward,

$$\text{IRIS}_{\text{ent}}(o_t|q, o_{<t}) := -\text{KL}(\pi_\theta(o_t|q, o_{<t})\|U) = \text{Entropy}(\pi_\theta(o_t|q, o_{<t})) - \log |U|,$$

where $|U|$ is the vocabulary size. Compared with minimizing the forward KL divergence with respect to the uniform distribution, minimizing the backward KL divergence is equivalent to maximizing the entropy. In Fig. 8, we show that backward KL divergence formulation is subpar to the forward counterpart, which is consistent with previous findings that self-certainty behaves better than entropy for the model's intrinsic self-certainty (Zhao et al., 2025b; Kang et al., 2025).

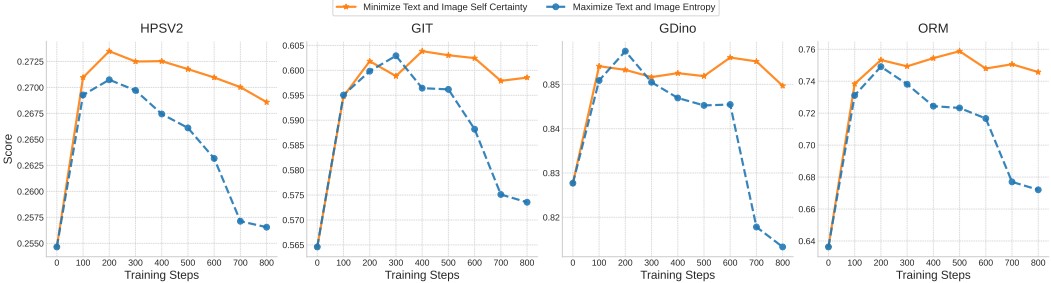

Figure 8: **Ablation study**: forward KL outperforms backward KL.

**Training with or without RL** The objective we aim to maximize $\text{NSC}(o_t|q, o_{<t}) = \text{KL}(U\|\pi_\theta(o_t|q, o_{<t}))$ is differentiable with respect to the model parameters. Therefore, it is natural to ask if it is necessary to adopt RL to maximize the objective, since we can directly maximize the NSC. The objective is then to maximize the following objective:

$$\mathcal{J}_{\text{NSC-direct}}(\theta) = \mathbb{E}_{\substack{q \sim P(Q), \\ O=\{o_i\}_{i=1}^G \sim \pi_{\theta_{\text{old}}}(\cdot|q)}} \frac{1}{G} \sum_{i=1}^{G} \frac{1}{|o_i|} \sum_{t=1}^{|o_i|} \left\{ \text{NSC}(o_{i,t}|q, o_{i,<t}) - \beta \text{KL}(\pi_\theta\|\pi_{\text{ref}}) \right\} ,$$

We adopt the same training configurations with our main experiments. In Fig. 9, we observe that directly maximizing the NSC will lead to model collapse. We explain this difference in the following: GRPO employs a more conservative strategy by first generating a batch of responses and then aligning with the largest NSC. However, directly optimizing the NSC might adopt a more aggressive update that may hurt the performance. Furthermore, aligning with prior work on intrinsic rewards (Zhao et al., 2025b; Zhang et al., 2025a), using RL to optimize a differentiable intrinsic reward is a standard practice.

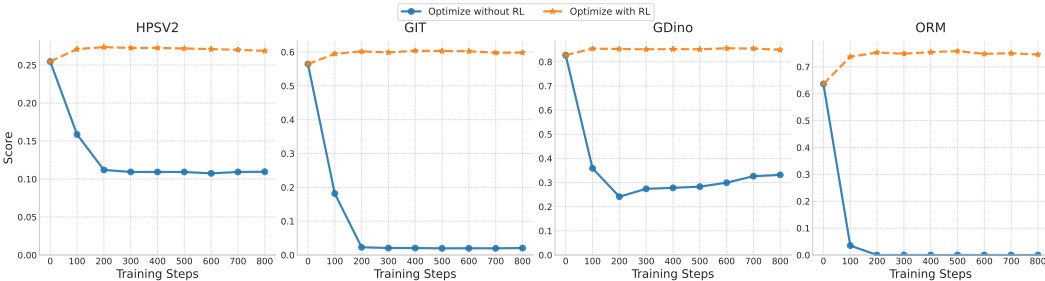

Figure 9: **Ablation study**: Optimizing with RL is more effective than directly maximizing the NSC.

## 4.4 Further discussions

In this work, we examine our intrinsic reward strategy, IRIS, on Janus-Pro, an autoregressive text-to-image model. While large language models have largely been dominated by the decoder-only architectures, the text-to-image models are far more diverse. They include a variety of competing model architectures such as continuous diffusion models (Zhou et al., 2024), masked-modeling approaches (Xie et al., 2024), and MAE-style models (Tong et al., 2024), with no single architecture having dominated. Therefore, exploring how intrinsic reward can be adapted and applied across these architectures is an interesting direction for future research.

## 5 Conclusion

In this paper, we proposed IRIS, a method that optimizes text-to-image models by leveraging negative self-certainty (NSC) as an intrinsic reward. Unlike RLHF or RLVF, our method doesn't need any human labeling or domain specific verifier, making it more scalable and easily generalizable to various domains. Our key intuition comes from the observation that less self-confident T2I models are more likely to generate visually rich and colorful images. Experiments demonstrate that IRIS achieves results comparable to verifiable external rewards, and even better results in the initial learning. Our work shows that contrary to the popular belief that higher self-certainty is generally beneficial for language reasoning, lower self-certainty encourages the generation of more visually rich and colorful images. Our work highlights the different roles of self-certainty in different modalities, offering a potential guideline for the development of future multimodal generative models.

## Ethics Statement

Our work on text-to-image generation is committed to responsible AI development and adheres to standard academic and ethical practices. We recognize the potential for misuse of text-to-image generation models, including the generation of misleading or harmful content such as deepfakes. This

particular project does not involve human subjects or raise concerns regarding data privacy, bias, or fairness in its current scope. Our research focuses on foundational architectural and training methodologies, with no direct application to the creation of sensitive or personally identifiable imagery. We are dedicated to ensuring that our research contributes to the safe and beneficial advancement of AI and are actively exploring methods to detect and prevent malicious applications of vision models.

## REPRODUCIBILITY STATEMENT

To ensure reproducibility of our results, we provide the following resources: (1) complete implementation details and hyperparameters are described in Sec. 4.1 and Appx. B.1; (2) all benchmarks and models used in our experiments are publicly available and properly cited with access information provided in Sec. 4.1 and Appx. A; and (3) source code will be made available upon publication to facilitate replication of our experimental results.

## THE USE OF LARGE LANGUAGE MODELS (LLMS)

We used a large language model (LLM) for assistance. Its primary roles were to aid in polishing the grammar and improving the style of the text throughout the paper. Following its generation, the authors carefully reviewed, edited, and rewrote the content to ensure its accuracy and alignment with the paper's standards. The authors take full responsibility for all content presented in this work.

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

# A    BENCHMARKS AND REWARD MODELS

In this section, we give a detailed description of the benchmarks and external reward models we used in the main paper.

## A.1    BENCHMARKS

**GenEval (Ghosh et al., 2023)**    GenEval is a object-centric framework for evaluating T2I models. First, we use the model to generate images based on testing prompts, which are divided into 6 categories: (1) *Single Object* (2) *Two Objects* (3) *Colors* (4) *Counting* (5) *Position* (6) *Color Attribution*. After image generation, we use an object detector (Chen et al., 2019; Cheng et al., 2022) to detect the targeted objects and CLIP ViT-L/14 to classify the object color. Each image receives a binary score indicating whether the described object is rendered correctly. Our evaluation is based on the 553 instructions in the GenEval's evaluation set. For each instruction, our model generates four candidate images, and we report the averaged score in one category.

**T2I-CompBench (Huang et al., 2023)**    T2I-CompBench is a compositional text-to-image generation framework for evaluating T2I models. For the attribute-binding task, we use disentangled BLIP-VQA (Li et al., 2022) on three attributes: *color*, *shape* and *texture*. For the object relationship, we use UniDet (Zhou et al., 2022) for *2D-spatial* relationship evaluation, and CLIPScore Radford et al. (2021) for *non-spatial* relationship evaluation. In summary, we use the we a 3-in-1 evaluation metric *complex* which computes the average score of CLIPScore, Disentangled BLIP-VQA, and UniDet, as the evaluation metric for complex compositions. Our evaluation is based on the 300 instructions in the T2I-CompBench's evaluation set of each attribute. For each instruction, our model generates four candidate images, and we report the averaged score of each attribute.

**WISE (Niu et al., 2025)**    WISE (World Knowledge-Informed Semantic Evaluation) is a comprehensive benchmark designed to evaluate the ability of T2I models to integrate and apply real-world knowledge beyond merely word-to-pixel matching. It consists of 1,000 prompts spanning three main categories: *cultural common sense*, *spatio-temporal reasoning*, and *natural science*. The spatio-temporal reasoning category is further divided into *time* and *space*, while natural science includes *biology*, *physics*, and *chemistry*. WISE introduces WiScore, a scoring metric that uses GPT-4o to quantify *consistency* (accuracy in depicting the prompt's content), *realism* (visual plausibility), and *aesthetic quality* (composition and visual appeal). We report the average score of each category.

**TIIF-Bench (Wei et al., 2025)**    TIIF-Bench (Text-to-Image Instruction Following Benchmark) is a comprehensive, and difficulty-graded benchmark designed to assess modern T2I models' ability to follow complex textual instructions, addressing limitations like simplistic prompts and coarse evaluation found in prior benchmarks. It has three major catgeries: *Basic Following* (*Attribute*, *Relation*, *Reasoning*), *Advanced Following* (*Attribute+Relation*,*Attribute+Reasoning*, *Relation+Reasoning*,*Style*,*Text*), and *Real World Following*. For each prompt, this benchmark contains the *short* and *long* versions to more systematically evaluate the model's ability to follow short and long instructions.

## A.2    REWARD MODELS

**Human Preference Model (HPSv2):**    To capture a generalized sense of image quality, we utilize a reward function derived from a Human Preference Model, HPSv2 (Wu et al., 2023). It is trained to learn human aesthetic preferences by learning from vast datasets of AI-generated images ranked by human annotators. When evaluating a new image, the model considers both its faithfulness to the text prompt and its overall visual appeal. These two factors are combined into a single score, which provides a comprehensive measure of the image's success.

**Object Detector (DINO):**    We employ an object detector, GroundingDINO (Liu et al., 2024) as a specialized "vision expert" to assess how accurately a generated image reflects the compositional elements of its prompt. This evaluation focuses on three key aspects: the existence of objects, their specified count, and their spatial relationships.

First, we parse the text prompt to create a target list of all mentioned objects $\{o_i\}_{i=1}^K$. The object detector then analyzes the generated image to locate these objects.

- **Spatial relationships**: If a prompt describes a spatial arrangement (e.g., "a cup to the left of a book"), we use the detected bounding boxes of the objects. We then calculate metrics like their relative distance and Intersection over Union (IoU) to produce a spatial accuracy score, $\mathcal{R}_{\text{spatial}}$.

- **Object count**: Otherwise, if the prompt specifies a particular number of an object, $n_{o_i}$, (e.g., "three cats"), we compare this target to the number detected by the model, $\hat{n}_{o_i}$.

- **Object existence**: Otherwise, for each of the $K$ target objects, we assign a binary score—1 if the object is detected in the image and 0 if it is not.

By combining these evaluations, the total reward from the object detector, $\mathcal{R}_{\text{Det}}$, is determined as:

$$
R_{\text{Det}} = \begin{cases} \alpha \mathcal{R}_{\text{spatial}} + (1-\alpha)\frac{1}{K}\sum_{i=1}^K \mathbb{I}(o_i \text{ detected}), & \text{if spatial relationship in the prompt,} \\ \frac{1}{n}\sum_{i=1}^K \mathbb{I}(n_{o_i} = \hat{n}_{o_i}), & \text{if number in the prompt,} \\ \frac{1}{n}\sum_{i=1}^K \mathbb{I}(o_i \text{ detected}), & \text{else,} \end{cases}
$$

where $\mathcal{R}_{\text{spatial}}$ is 1 if the relative distance between the objects is larger than a threshold and the direction is right. If the direction is wrong, the reward is 0. Otherwise, we use the IoU as the spatial reward. We set $\alpha$ as 0.6 to encourage the correctness of the spatial relationship.

**Visual Question Answering Model (GIT):**  We employ a Visual Question Answering (VQA) model, GIT (Wang et al., 2022), to assess the presence and attributes of objects in generated images by answering image-related questions. The model is trained on question–answer pairs derived from visual content.

Our methodology involves transforming the image prompt into a series of targeted questions. For example, a prompt such as *a blue bird and a red horse* is decomposed into individual queries like "*Is there a blue bird?*" and "*Is there a red horse?*". For each query $i$, we extract the model's output probabilities for the answers "Yes" ($P_{\text{Yes}}^i$) and "No" ($P_{\text{No}}^i$).

The final reward score, $R_{\text{VQA}}$, is computed by averaging the normalized probability of a "Yes" answer over all $K$ queries derived from the prompt. This is formally defined as:

$$
R_{\text{VQA}} = \frac{1}{K}\sum_{i=1}^K \frac{P_{\text{Yes}}^i}{P_{\text{Yes}}^i + P_{\text{No}}^i}.
$$

**Output Reward Model (ORM):**  We incorporate an Output Reward Model (ORM) (Guo et al., 2025b) to provide an assessment of complete prompt-image alignment. The ORM is a Large Multimodal Model (LMM), such as LLaVA-OneVision (Li et al., 2024a), that has been specifically fine-tuned for this purpose. The fine-tuning objective instructs the model to act as a binary evaluator, outputting "Yes" only if the generated image perfectly aligns with the entire text prompt, and "No" otherwise.

The calculation of the reward, $R_{\text{ORM}}$, is similar to the VQA-based reward. The primary difference is that we provide the complete, original prompt to the ORM as a single query rather than decomposing it. The reward is thus the normalized probability of the model returning a "Yes" response for the complete prompt-image pair:

$$
R_{\text{ORM}} = \frac{P_{\text{Yes}}}{P_{\text{Yes}} + P_{\text{No}}}.
$$

# B  ADDITIONAL EXPERIMENTS DETAILS

## B.1  EXTERNAL REWARD TRAINING INFLUENCE SELF-CERTAINTY

We give the experiment details omitted in Sec. 1 and Fig. 2.

**Image generation model**  We adopt Janus-Pro-1B (Chen et al., 2025) and Janus-Pro-7B as our image generation model. We finetune it for 1000 steps, using Generative Reward Process Optimization (GRPO) (Shao et al., 2024). We use the sum of the four external rewards described in Appx. A.2. Training is distributed across 4 NVIDIA A6000 GPUs, managed by `torchrun` and optimized with DeepSpeed's ZeRO stage 3, `bfloat16` mixed precision, and FlashAttention-2.

**Text generation model**  We adopt Qwen2.5-1.5B-Instruct (Yang et al., 2024a) as our text model. We finetune it for 5000 steps on MATH-lighteval (Hendrycks et al., 2021), using Generative Reward Process Optimization (GRPO) (Shao et al., 2024). We use accuracy, format and tag count as our external rewards (Shao et al., 2024). Training is distributed across 4 NVIDIA A6000 GPUs, managed by `torchrun` and optimized with DeepSpeed's ZeRO stage 3, `bfloat16` mixed precision, and FlashAttention-2.

### B.2  Results on Janus-Pro-7B

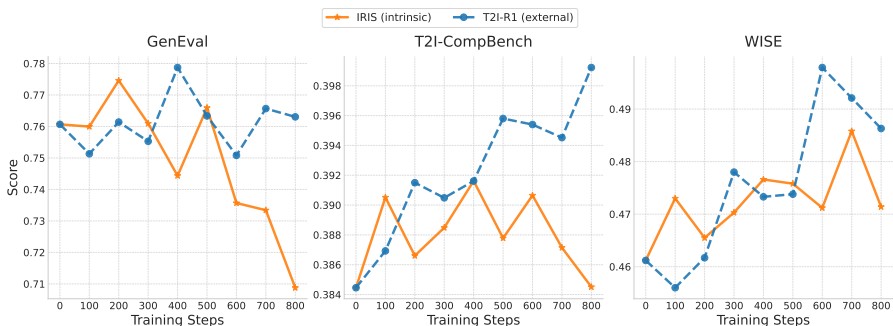

Figure 10: **Main results of Janus-Pro-7B** on GenEval, T2I-CompBench and WISE.

### B.3  Sub-category scores in benchmarks

We give the sub-category score of three different benchmarks in Fig. 11 and Fig. 12. We discover that IRIS is comparable in the T2I-CompBench and WISE benchmarks. More over, we discover that RL finetuning will boost the 1B-sized model more than the 7B-sized model.

### B.4  Textual generation results

Fig. 13 illustrates that semantic CoTs generated by the model trained with intrinsic rewards at the step 0 (the base model), step 400, and step 800. We find that RL finetuning on intrinsic rewards can incentivize the emergence of long-form reasoning.

### B.5  Human Evaluation

For the human evaluation, we randomly sample 100 prompts from the three benchmarks (WISE, T2I-Compbench, GenEval). Due to limited resources, we only evaluated three checkpoints: (1) **Janus-Pro-1B** (base generation model), (2) **T2I-R1 (external reward)**: Janus-Pro-1B finetuned with four external rewards, and (3) **IRIS (internal reward)**: Janus-Pro-1B finetuned with our intrinsic reward. For (2) and (3), we use the best checkpoints selected on WISE. We then ask 9 human evaluators to choose the best image among the three for each prompt and record the number of times each model is upvoted. The results are reported in Tab. 2. Our method achieves performance comparable to the external-reward-based model.

### B.6  Image generation results

In Fig. 14, we present some examples in GenEval, T2I-CompBench and WISE. The base model is Janus-Pro 1B. We find that IRIS could improve the image generation ability.

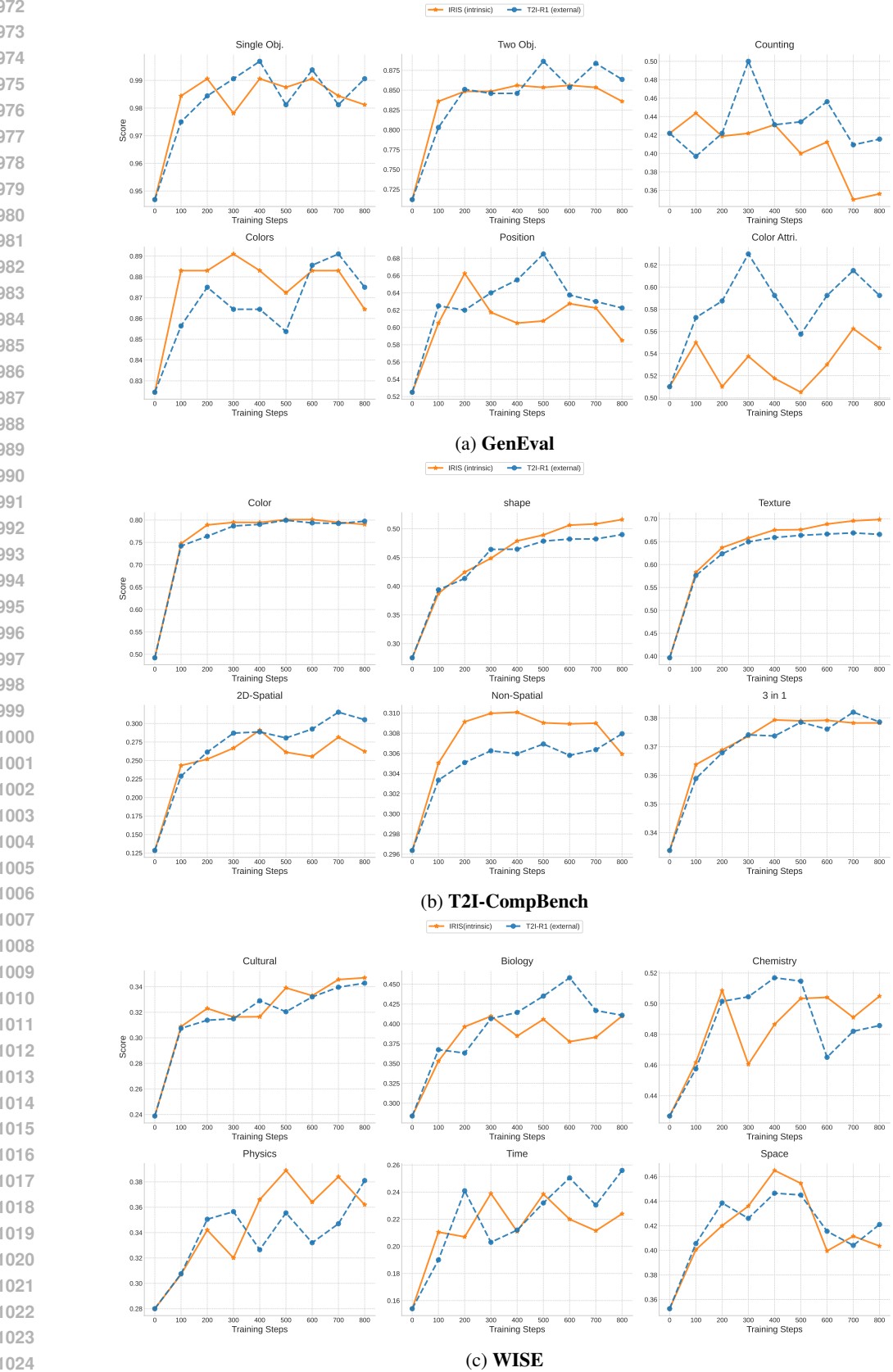

Figure 11: Subscores of Janus-Pro-1B on GenEval, T2I-CompBench and WISE

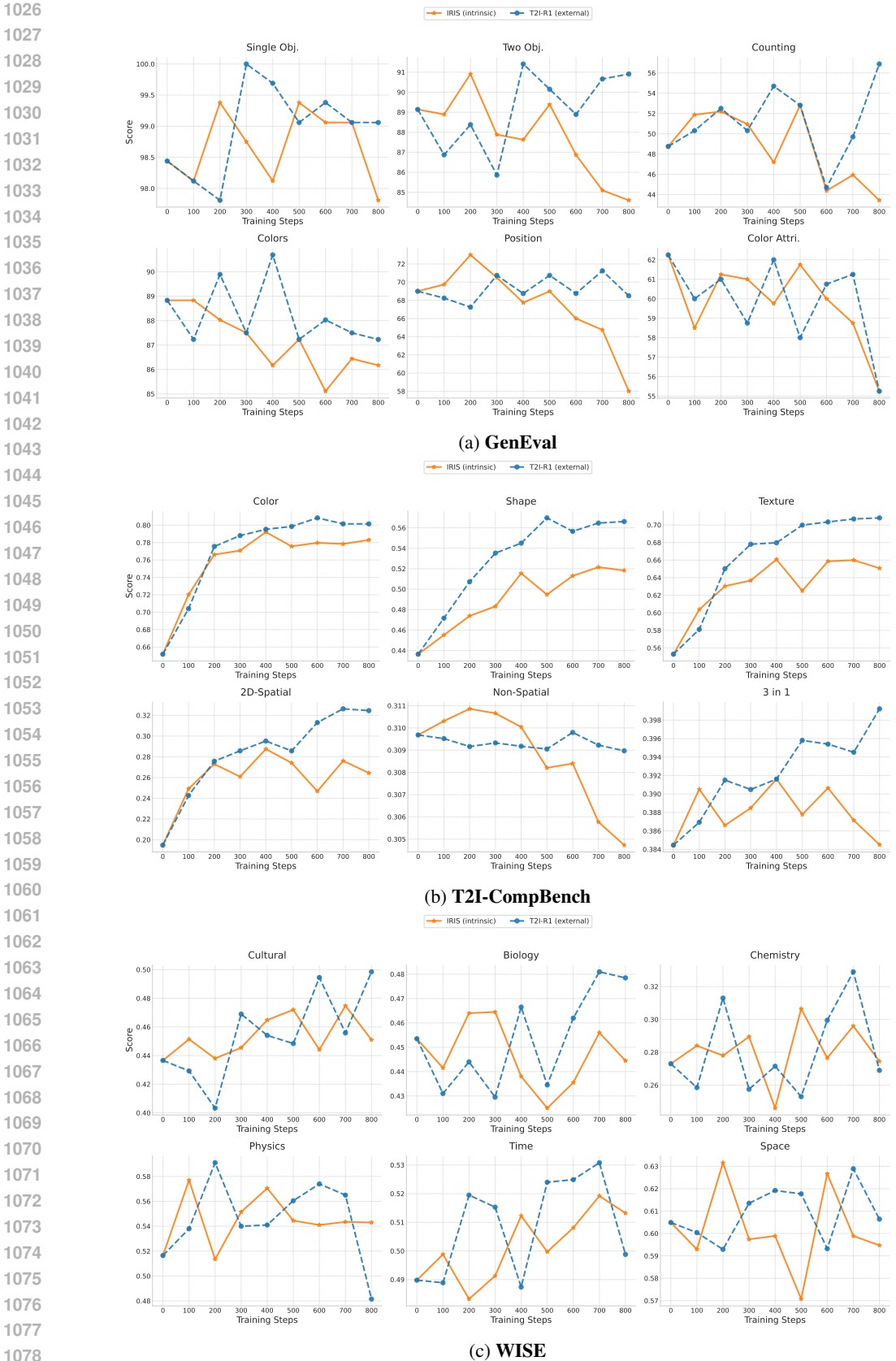

Figure 12: Subscores of Janus-Pro-7B on GenEval, T2I-CompBench and WISE

**Single object**

**Prompt:** a photo of a computer keyboard

**Step 0:** Here is a 50-word, concise visualization of the prompt: A laptop keyboard in black and white.

**Step 400:** A computer keyboard with a white keyboard, green keys labeled 'A' through 'Z', blue arrow keys with numbers from '0' to '9', and functional keys arranged in a standard QWERTY layout.

**Step 800:** Here is a visualization of a photo of a computer keyboard: A photo of a computer keyboard in the foreground, a white screen in the background, and a black keys with numeric keys on the right-hand side. The keys have a black background, and the numbers on them are blue with thick black borders. The screen is white with a faint grid pattern in the middle. Space is arranged uniformly around the keyboard, giving the image a clean and organized appearance. The keys on the keyboard are arranged in a logical pattern of function with the arrow keys highlighted. No extraneous text is present in this image.

**Color attribution**

**Prompt:** a photo of an orange traffic light and a white toilet

**Step 0:** [Image of an orange traffic light with "STOP" written on it] [Image of a white toilet with the label "TOILET" written on it]

**Step 400:** A close-up photo showcasing an orange traffic light with a white toilet in the background. The orange traffic light features a reflective surface with a glossy sheen, the toilet has a smooth, ceramic texture with a typical white finish, placed in front of a wooden fence.

**Step 800:** The image shows an orange traffic light and a white toilet, with the traffic light at the center and the toilet positioned to the top right side of the image. The traffic light is rectangular with orange stripes and a lit bulb, and the toilet is round with white walls, a white seat, and a large opening. The contrast between the bright orange and white colors creates a striking visual, with the traffic light standing out as the primary object while adding a touch of humor to the mundane scene.

**Position**

**Prompt:** a photo of a couch below a potted plant

**Step 0:** Here is a 50-word, concise visualization of the prompt: A photo of a couch below a potted plant with a gradient background transitioning from green to blue.

**Step 400:** A cozy-looking couch with light brown and beige tones is adorned below a small green potted plant with smooth, rounded leaves. The couch features a cushiony fabric, and the plant's vibrant green complements its natural appearance. The scene is vibrant and lively, set in a bright, sunlit room.

**Step 800:** The image shows a couch and a potted plant positioned underneath one another, both in a living room setting. The color of both elements is green, the couch is light brown in color with a wood texture, and the plant has a vibrant green color without any visible leaves. Shape-wise, the couch is rectangular, while the potted plant is round with a leafy texture. There is a faint background of neutral colors, emphasizing the contrast between the couch and the plant. The relationship between the elements is that the plant is in an angled position in front of the couch, complementing the shape and placement of the sitting furniture.

Figure 13: Examples of semantic CoTs given the prompts in the three categories: *single object*, *color attribution* and *position*.

| Method | Janus-Pro-1B | T2I-R1 External reward | IRIS (Internal reward) |
|---|---|---|---|
| Rate | 0.13 | 0.42 | 0.45 |

Table 2: Human evaluation results

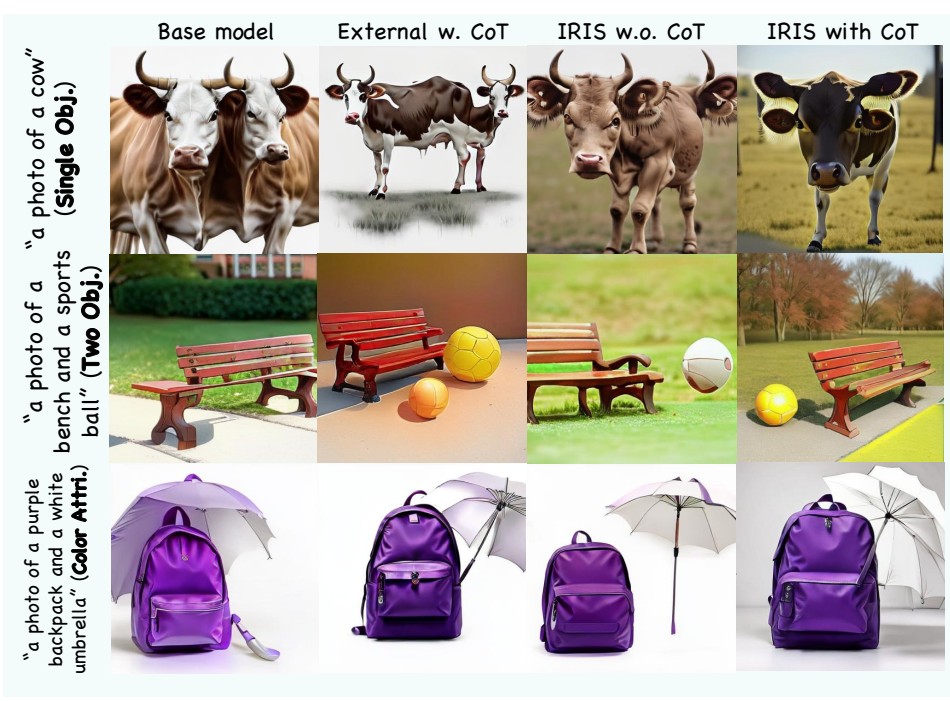

(a) GenEval.

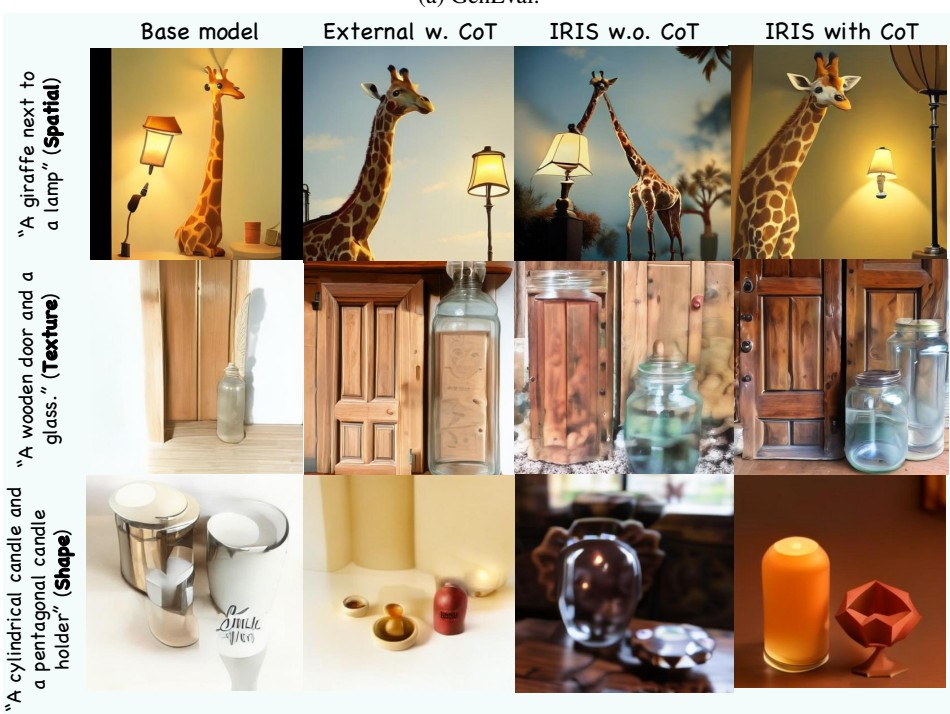

(b) T2I-CompBench.

Figure 14: Examples of generated images (part 1).

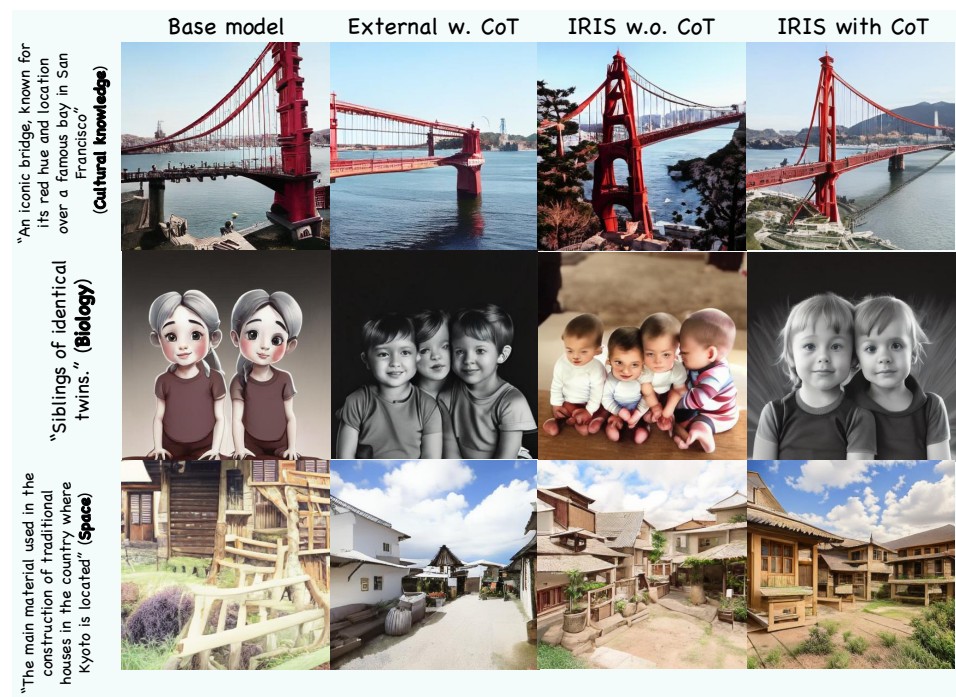

(c) WISE.

Figure 14: Examples of generated images (part 2).

## B.7 TIIF-BENCH

We conduct additional experiments on TIIF-Bench, and present results in Tab. 3. We use the testmini prompts, but run each prompt 4 times to average the performance.

Table 3: "Und." and "Gen." denote "understanding" and "generation", respectively. We report the scores of the last checkpoint of the T2I-R1 and IRIS.

| Type | Model | Overall | | Basic Following | | | | | | | | Advanced Following | | | | | | | | | | | | Designer | |
| | | | | | Avg | | Attribute | | Relation | | Reasoning | | Avg | | Attribute +Relation | | Attribute +Reasoning | | Relation +Reasoning | | Style | | Text | | Real World | |
| | | short | long | short | long | short | long | short | long | short | long | short | long | short | long | short | long | short | long | short | long | short | long | short | long | short | long |
| Gen. Only | PixArt-α (Chen et al., 2023b) | 44.37 | 50.50 | 55.50 | 61.00 | 52.33 | 56.33 | 63.82 | 74.07 | 50.32 | 52.57 | 38.71 | 44.90 | 37.82 | 41.32 | 58.84 | 52.46 | 40.22 | 47.09 | 50.00 | 76.67 | 0.00 | 0.83 | 45.70 | 53.16 |
| | SDXL (Podell et al., 2023) | 54.96 | 42.13 | 65.72 | 53.28 | 59.33 | 50.83 | 77.57 | 62.57 | 60.32 | 46.57 | 49.73 | 36.22 | 47.82 | 35.57 | 56.22 | 45.34 | 52.59 | 36.09 | 73.33 | 60.00 | 16.83 | 0.83 | 50.92 | 41.59 |
| | FLUX.1 (Labs, 2024) | 71.09 | 71.78 | 83.12 | 78.65 | 87.05 | 83.17 | 87.25 | 80.39 | 75.01 | 72.39 | 65.79 | 68.54 | 67.07 | 73.69 | 73.84 | 73.34 | 69.09 | 71.59 | 66.67 | 66.67 | 43.83 | 52.83 | 70.72 | 71.47 |
| | SD3-Medium (Esser et al., 2024a) | 70.17 | 66.19 | 80.20 | 75.20 | 84.50 | 77.00 | 78.90 | 77.51 | 77.21 | 71.08 | 62.33 | 66.49 | 66.54 | 77.51 | 57.92 | 62.32 | 61.46 | 63.53 | 80.00 | 73.33 | 53.39 | 28.51 | 71.64 | 64.93 |
| Und. & Gen. | Show-o (Xie et al., 2024) | 59.72 | 58.86 | 73.08 | 75.83 | 74.83 | 79.83 | 78.82 | 78.32 | 65.57 | 69.32 | 53.67 | 50.38 | 60.95 | 56.82 | 68.59 | 68.96 | 66.46 | 56.22 | 63.33 | 66.67 | 3.83 | 2.83 | 55.02 | 50.92 |
| | Janus-Pro-1B (Chen et al., 2025) | 61.29 | 60.91 | 66.56 | 71.32 | 67.38 | 70.62 | 68.68 | 78.94 | 63.63 | 64.39 | 53.10 | 60.15 | 59.23 | 64.08 | 53.61 | 62.66 | 54.69 | 62.30 | 22.50 | 38.33 | 17.99 | 13.12 | 61.29 | 60.91 |
| | Janus-Pro-1B + T2I-R1 (Jiang et al., 2025) | 64.18 | 69.50 | 73.41 | 73.19 | 72.38 | 70.38 | 79.38 | 76.98 | 68.47 | 72.21 | 61.79 | 64.07 | 68.09 | 67.20 | 60.39 | 67.38 | 66.76 | 67.28 | 23.33 | 36.67 | 21.27 | 14.14 | 64.18 | 69.50 |
| | Janus-Pro-1B + IRIS (Ours) | 69.64 | 66.42 | 73.91 | 74.46 | 74.81 | 75.00 | 79.45 | 77.67 | 67.45 | 70.71 | 60.03 | 63.84 | 66.57 | 69.97 | 59.02 | 68.00 | 63.54 | 62.89 | 28.33 | 40.00 | 19.40 | 12.90 | 69.64 | 66.42 |

## B.8 CONTINUOUS TRAINING RESULTS

We perform experiments to examine whether IRIS can continuously obtain performance gains throughout the training process. To this end, we train IRIS on Janus-Pro-1B for 1600 steps and evaluate the checkpoints on the GenEval benchmark. The results are shown in Fig. 15. Intrinsic rewards substantially improve performance in the early stages of training, but the performance gradually degrades as training progresses, which is also reported by Zhang et al. (2025b).

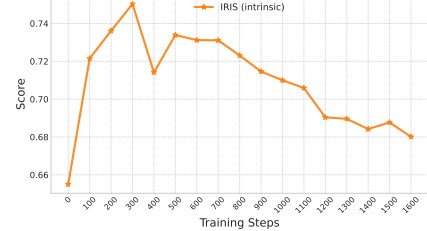

Figure 15: Continuing training results. The performance will drop when continuing training, but still higher than the base model.

## C LIMITATIONS

We discuss several limitations in this work.

**Autoregressive models**  Our method is currently limited to autoregressive (AR) text-to-image models. The primary reason is that its foundation in RL requires calculating the next-token probability distribution. Applying RL to other types of text-to-image models, such as flow matching or masked models, is challenging. AR models simplify this because they directly model the per-token log-probabilities. Consequently, the sequence-level log-probability $\left(\log \pi_{\mathrm{AR}}(o|q) = \sum_{i=1}^{t} \log \pi_{\mathrm{AR}}(o_t|q, o_{t-1})\right)$ can be easily computed through the chain rule using a single forward pass. Recently, Liu et al. (2025) proposed Flow-GRPO, which first integrates policy gradient RL into flow matching models. However, these models do not operate on discrete tokens like Janus-Pro or Skywork-UniPic. Instead, their update rule involves continuous values, as seen in $x_{t+\delta t} - x_t = \mu(x_t, t)\delta t + \mathcal{N}(0, \sigma_t\sqrt{\delta t})$. This means the policy is an isotropic Gaussian distribution with fixed $\sigma_t$, and its forward KL divergence with respect to the uniform distribution is a non-parameterized constant that cannot be optimized. Furthermore, masked text-to-image models lack an autoregressive structure and thus do not have the sequential factorization of the sequence log-probability. Therefore, current AR-based RL algorithms, such as GRPO, cannot be directly applied to them. Addressing these models with RL, as seen in works like Liu et al. (2025) and Zhao et al. (2025a), is a research direction separate from our current focus.

**Longer training performance**  We observe that the performance of the model may drop after extended training. A similar degradation phenomenon has been reported in Zhang et al. (2025b), where the authors claim that RLIF can initially boost reasoning performance, matching or surpassing RLVR, but the performance degrades as training progresses. Furthermore, previous works on RLIF, such as Zhao et al. (2025b), typically only report early-stage performance, training for only 60 steps.

