# OpenReview forum: "IRIS: Intrinsic Reward Image Synthesis"
_ICLR.cc/2026/Conference — Submitted to ICLR 2026_

### Official Review · Reviewer_kaX6 · 2025-10-30

**Soundness:** 3
**Presentation:** 3
**Contribution:** 2
**Rating:** 2
**Confidence:** 4

**Summary:**

This paper introduces IRIS, which is a RL reward based on minimizing self-confidence / maximizing self uncertainty. The authors propose this objective and use GRPO to train an autoregressive text to image model. The performance of the IRIS model is on par with other methods which use a reward signal for training.

**Strengths:**

- This paper is well written and does a good job of communicating the idea
- The ablations are good.
- I appreciate the novelty, and I appreciate the insight that encouraging mode covering behavior in autogregressive image generators trends towards human preferences.

**Weaknesses:**

- This approach seems limited to optimizing for human preferences as measured by these specific benchmarks. If for example, I wanted to fit to the preference of some specific group, then this method would not work.
- The motivation for choosing this method over alternatives isn't entirely clear to me. For instance, would the approach remain effective with a different base model than Janus Pro? Would this generalize better to other human preference benchmarks?
- I'd appreciate more understanding into the underlying mechanisms of this behavior. Is this an emergent property of the pretraining phase, or something else?
- I am surprised that figure 2 allows one to draw this conclusion. For example, [0] and other papers indeed observe an increase in entropy in addition to an increase in reward. I notice that the GRPO for the LLM is on a MATH task, which entropy does usually decrease.

- Unimportant but appendix b.1 refers to GRPO as Generative Reward Process Optimization, instead of Group Relative Policy Optimization.

[0] Tonyi Deep Research, https://arxiv.org/pdf/2510.24701

**Questions:**

- is IRIS more robust to multiple benchmarks? why would i prefer iris over other methods?
- what is the hypothesis for this working?
- does this method work for other families of models?

---

> ### Author Response · Authors · 2025-11-21
>
> Thank you for providing your valuable feedback. We have answered your questions in the following and updated the paper accordingly. We hope our resposes have addressed your concerns.
>
> 1. **Human preference survey** (Weakness 1)
>
> See **Common Response: Human preference survey** and **Common Response: Additional benchmarks**. We conduct additional experiments on human study and TIIF-Bench, both of which favor our IRIS. This verifies the generalization capacity of our method, and we are not focusing on the persoalized generation models in particular.
>
> 2. **Why prefer IRIS** (Question 1)
>
> We have added an additional benchmark TIIF-Bench. See **Common Response: Additional benchmarks**. We propose IRIS to show that using only model's intrinsic signals can match the performance of relying on current external models. Our goal is not simply to outperform methods using external rewards, but to study the role of model confidence and to call for better external reward models that can capture features beyond the capabilities of intrinsic model signals. See **Common Response: Motivation and contribution**.
>
>
> 3. **Beyond auto-regressive models.** (Weakness 2, Question 3)
>
> See **Common Response: Beyond auto-regressive models**.
>
> 4. **Mechanism** (Weakness 3, Question 2)
>
> We believe the mechanism should be explained in the post-training stage. Different tasks requires the model to have different levels of self-certainty: higher self-certainty benefits the model in domains requiring objective reasoning (e.g., mathematical and code reasoning), but lower self-certainty benefits the model in subjective generation tasks (e.g., text-to-image generation, creative generation). Specially, our work is inspired by the finding that *increased self-certainty typically results in more uniform and less visually diverse images, while lower self-certainty tends to generate images with richer visual features that are more preferred by humans* (see Fig. 1).
>
> 5. **Increase in entropy on text tasks** (Weakness 4)
>
> Thanks for pointing it out. We have revised our discussion in Section 2. Kindly notice that this paper is release on Oct 28 2025, after the ICLR deadline. Their findings in Figure 8 coincide with our algorithn, where we also maximize the NSC on the text tokens, i.e., increase uncertainty or maximize entropy. (See **Common Response: Reward aggregation details**). In the text-to-image task where IRIS is built on, the model generates detailed text description first to help the image generation. Therefore, text is explorative here. And in the information-seeking seeking tasks in [1], text is also explorative. In both settings, uncertainty will encourage the exploration in text generation and can be beneficial to the performance, in contrast with math/code reasoning where certainty and accuracy is required. We have added this discussion in Section 3.2.
>
> 6. **Typos** (Weakness 5)
>
> Thank you for pointing out the typo. We have corrected it in the updated version.
>
> [1] Tongyi DeepResearch Team, Baixuan Li, Bo Zhang, Dingchu Zhang, Fei Huang, Guangyu Li, Guoxin Chen, Huifeng Yin, Jialong Wu, Jingren Zhou, et al. Tongyi deepresearch technical report. arXiv preprint arXiv:2510.24701, 2025.

---

> ### Comment · Reviewer_kaX6 · 2025-11-26
>
> I thank the authors for their reply. But so far, I would maintain my score for the following reasons:
>
> (1) The argument that reward models are not doing enough relative to just maximizing uncertainty is valid but I feel this is not a sufficient contribution yet to warrant a full conference paper. I believe that progress towards the next step after this observation would be required.
>
> (2) To make such a claim, one needs to ablate over other models and algorithms, but this is not done. Ie: how do we know that this is a failure of the reward model instead of a failure of the algorithm?
>
> (3) I would like to see further experiments to show this claim. For example, if NSC maximization is good for these tasks, perhaps one could show that there is high preference alignment between using NSC and the preference reward models? This would be much more convincing than the current paper.

---

> ### Author Response · Authors · 2025-11-27
>
> Thanks for your response. We understand your concern. We address your concern in the following way.
>
> (1) We argue that maximizing self-uncertainty is a sufficiently novel contribution. First, this approach stands in direct contrast to the Reinforcement Learning with Internal Feedback (RLIF) literature, where the objective is typically to maximize self-certainty. While prior work shows that increasing model certainty can improve language model performance on math and code tasks, we find that in the text-to-image generation domain, decreasing model certainty leads to more diverse and, consequently, higher-quality images. Thus, our objective is fundamentally different. Existing work focuses on enforcing internal consistency in the model’s outputs, whereas our goal is to encourage exploration. This defines a new regime in the RLIF literature that has not been previously explored. Moreover, our findings highlight a new question for RLIF: different tasks and domains may require different optimal levels of model confidence. This insight might also apply to the pretraining stage, prompting a rethinking of how to adapt exploration levels across different kinds of tasks.
>
> (2) We explained in the **Common Response: Beyond auto-regressive models** that the current RL finetuning on text-to-image models can be applied to autoregressive text-to-image models. Regarding your second question, We admit that, the observation that IRIS can compete with external rewards can also be explained by the assumption that, RL algorithm (GRPO in our paper) cannot learn the signals in the external reward models, so it may not be owing to the external reward models' incapabilities. Following your suggestion, we have modified the Contribution 3 in the following way.
> ```
> Empirically, we show that IRIS can improve the autoregressive T2I models without any external supervision. Post-training by IRIS achieves competitive performance compared to training with external rewards. We call for better design of training algorithm and external reward models that are beyond the intrinsic capabilities of the base T2I model.
> ```
> (3) To show the alignment of the NSC and the external (preference) reward models, we adopt the following method:
>
> - External reward maximization: We directly maximize the external reward models using Reinforcement Learning (RL). During this training, we monitored the average NSC, expecting it to increase if the models are aligned. As shown in Figure 2, we observe a clear positive correlation, where maximizing the external reward model leads to a increase in the NSC.
>
> - NSC Maximization: Conversely, we directly maximize the NSC using Reinforcement Learning (RL). We then monitored the external rewards generated. The results, presented as the orange line in the ablation study (Figure 5), demonstrate that maximizing the NSC also leads to an increase in the external reward score.

---

### Official Review · Reviewer_TNZX · 2025-10-31

**Soundness:** 1
**Presentation:** 2
**Contribution:** 2
**Rating:** 2
**Confidence:** 4

**Summary:**

This paper proposes IRIS, a label-free RL fine-tuning scheme for text-to-image models (Janus-Pro) that replaces external reward models with an intrinsic token-level signal called Negative Self-Certainty (NSC), defined as the forward KL from a uniform distribution to the model’s next-token distribution; the authors consider both text and image tokens and ultimately use NSC for both. The reward is optimized with a GRPO objective. Experiments compare three reward schemes—external, SC, and NSC—on Janus-Pro and evaluate early training (first ~800 steps) on GenEval, T2I-CompBench, and WISE, where IRIS reports gains over the base models and competitive performance with the external-reward baseline.

**Strengths:**

- IRIS uses an intrinsic reward (their NSC) instead of any human labels or domain-specific verifiers, which makes the setup lightweight and potentially scalable to new domains.
- The GRPO objective and token-level advantage computation are written explicitly, which lowers the barrier for reproduction or adaptation.
- The plots in the experimental results consistently report improvements early in training.

**Weaknesses:**

- This paper’s statements about self-confidence are mistaken. As written, IRIS is actually maximizing self-confidence, not minimizing it. The paper’s prose repeatedly says the opposite, but the mathematics and the reward used in training point the other way. Eq. (2) defines Self-Certainty (SC) at a token as $SC = -{KL}(U\ \Vert\ \pi_\theta)$, contradicting the typical definition [r1].
- There is a severe mischaracterization of the proposed method IRIS. The papers asserts that forward KL to uniform is “mode-covering,” contrasting it with entropy; but with the paper’s own definition (forward KL: ${KL}(U\Vert \pi)$), increasing NSC pushes peaky (confidence-seeking) distributions, not mode-covering.
- Metric terminology drifts mid-paper. Fig. 2’s paragraph says they “compute the self-confidence measured by ${KL}(U\Vert \pi)$,” which is NSC by their Eq. (2), while elsewhere “self-certainty” is plotted and discussed, resulting in inconsistent naming (SC vs. NSC vs. “confidence”).
- Novelty relative to prior work (INTUITOR [r1]) is thin at the algorithmic core. IRIS’s intrinsic reward is exactly the token-level forward KL to uniform, used inside GRPO—the same scalar that INTUITOR optimizes (called “self-certainty” there) using the same GRPO algorithm. The contribution is primarily domain/application (T2I) and pipeline choices (semantic CoT), not a novel objective. This should be explicitly acknowledged and compared.
- Gains in the experimental results may stem from the semantic CoT stage rather than the intrinsic reward per se. While the appendix shows with/without CoT comparisons, the main results and claims still bundle CoT with IRIS; stronger controls are needed (e.g., CoT-matched baselines and cross-combinations with external-reward pipelines).
- Although IRIS trains without external verifiers, most metrics are pretrained reward/evaluator models (HPSv2, etc). Improvements against these may reflect alignment with those evaluators rather than human preference; explicit human-preference studies can only be implied with blinded A/B tests.

[r1] Zhao, X., Kang, Z., Feng, A., Levine, S. and Song, D., 2025. Learning to reason without external rewards. arXiv preprint arXiv:2505.19590.

**Questions:**

- What exactly is “self-confidence” in Fig. 2? The text says you “compute the self-confidence measured by KL between uniform and the model’s distribution.” Is this NSC (i.e., ${KL}(U\|\pi)$)? If so, please clarify terminology and axes so readers don’t confuse SC with NSC.
- Why NSC on both text and image tokens? You conclude “using NSC as the intrinsic reward in both text and image tokens can achieve the best results.” Could you share ablations with mixed choices (e.g., NSC on image, SC on text), modality-specific weights, or temperature controls to support this decision?
- Many plots focus on the first ~800 steps. Do your gains persist or change at longer horizons, and how does variance across random seeds look over full training? Please include long-run curves and seed-wise spreads.
- The reward NSC is differentiable against the model parameters as it’s an analytic function of the model’s softmax probabilities, hence smoothly differentiable w.r.t. the logits/parameters. It means that it can be direclty optimized using gradient-based algorithm like SGD. What is the meaning of using RL to optimize it?

---

> ### Author Response · Authors · 2025-11-21
>
> Thank you for providing your valuable feedback. We have answered your questions in the following and updated the paper accordingly. We hope our resposes have addressed your concerns.
>
> 1. **Definition of the SC and NSC** (Weakness 1 and Question 1)
>
> Sorry for the confusion. There is a typo in Equation (2) ${\rm SC}(o_t|q,o_{<t}) := +{\rm KL}(U\|\pi_{\theta}(o_{t}|q,o_{<t}))$ is the correct definition. Thus, maximizing SC corresponds to maximizing the forward KL divergence from the uniform distribution, encouraging \(\pi_{\theta}\) to deviate from uniform and become more peaked, i.e., to have higher confidence/certainty. Therefore, our objective to maximize, NSC, is the exactly opposite objective, SC, in Intuitor [1]. This definition is consistent with our discussion throughout the paper. Regarding the link to [1], we have discussed it in Line 194-196.
> ```
> This is in clear contrast
> to previous works on pure text generation models, which maximize SC (Zhao et al., 2025; Zhang
> et al., 2025). In conclusion, we use NSC as our intrinsic reward in IRIS.
> ```
>
> 2. **Mode covering or mode seeking** (Weakness 2)
>
> We put the learnable distribution $\pi$ on the second term of the KL divergence. We maximize the NSC, by definition is minimizing the KL divergence to uniform distribution. Minimizng the the forward KL will encourage the mode-covering instead of the mode seeking, which aligns with our target to encourage exploration, [see here](https://www.tuananhle.co.uk/notes/reverse-forward-kl.html).
>
> 3. **Terminology Drifts** (Weakness 3)
>
> Sorry for the confusion. We use “self-certainty” and “self-confidence” to refer to the same concept, which caused the inconsistent terminology (SC vs. NSC vs. “confidence”) in the paper. We have carefully checked the manuscript and now use “self-certainty” consistently throughout.
>
> 4. **Limited novelty** (Weakness 4)
>
> We apologize for the confusion regarding NSC and SC. Our method optimizes an objective opposite to that of Intuitor. Specifically, our approach is based on the observation that *increased self-certainty typically results in more uniform and less visually diverse images, while lower self-certainty tends to generate images with richer visual features that are more preferred by humans* (see Fig. 1). This is clearly distinguished from Intuitor, which claims that *optimizing for confidence should improve reasoning capabilities.*
>
> We believe IRIS can contribute to the community, as we observe and confirm that a model’s self-certainty exhibits task-dependent behavior: self-certainty tends to increase in domains requiring objective reasoning (e.g., mathematical and code reasoning), but decreases in subjective generation tasks (e.g., text-to-image generation).
>
> 5. **Semantic CoTs** (Weakness 5)
>
> Sorry for the confunsion. Unless specified otherwise in the ablation study, all models in our paper is trained and evaluated with semantic CoTs.
>
>
> 6. **Human preference survey** (Weakness 6)
>
> See **Common Response: Human preference survey**.
>
> 7. **Reward aggregation details** (Question 2)
>
> See **Common Response: Reward aggregation details** In addition, we conduct experiments detailed in Section 4.3 to study whether minimizing or maximizing NSC on image/text tokens leads to better text-to-image (T2I) performance. Please refer to Figure 6 and Figure 7 for more details. A short conclusion here is that maxmizing both text and image NSC lead to better results. The reason might be lower self-certainty benefits the model in subjective generation tasks and encourages exploration (e.g., text-to-image generation).

---

> ### Author Response · Authors · 2025-11-21
>
> 8. **More steps** (Question 3)
>
> We observe that the performance of the model may drop after extended training. A similar degradation phenomenon has been reported in \cite{zhang2025no}, where the authors claim that RLIF can initially boost reasoning performance, matching or surpassing RLVR, but the performance degrades as training progresses. Furthermore, previous works on RLIF, such as \cite{zhao2025learning}, typically only report early-stage performance, training for only 60 steps. This degradation may be expected, as the intrinsic reward does not rely on any external knowledge. We emphasize again that our goal is not to outperform methods using external rewards, but rather to shed light on the roll of self-certain in modern generation models and to call for the development of better reward models.
>
> 9. **Directly optimizing NSC** (Question 4)
>
> We directly maximizing the NSC by maximizing
> $$
>  \mathcal{J}_{\text{NSC-direct}}(\theta) =  \mathbb{E}_{{
>      q \sim P(Q),\\ O=\{o_i\}_{i=1}^G \sim \pi_{\theta_{\text{old}}}(\cdot|q)}}
> \frac{1}{G}\sum_{i=1}^G \frac{1}{|o_i|}\sum_{t=1}^{|o_i|}\left\{
> \operatorname{NSC}(o_{i,t}|q,o_{i,<t}) -\beta\mathrm{KL}\bigl(\pi_\theta\Vert\pi_{\mathrm{ref}}\bigr)\right\}\,.
> $$
>
> Our experiments demonstrate that directly optimizing the NSC does not perform as well as using RL (Figure 9 in the updated paper). We explain this difference by: GRPO employs a more conservative strategy by first generating a batch of responses and then aligning with the best-performing output. In contrast, directly optimizing NSC can lead to more aggressive updates that may distort the probability structure of the trajectory. The detailed experimental results supporting this conclusion are provided in our ablation study. Furthermore, aligning with prior work on intrinsic rewards [1, 2], using RL to optimize a differentiable intrinsic reward is a standard practice.
>
>
> [1] Xuandong Zhao, Zhewei Kang, Aosong Feng, Sergey Levine, and Dawn Song. Learning to reason without external rewards. arXiv preprint arXiv:2505.19590, 2025b.
>
> [2] Qingyang Zhang, Haitao Wu, Changqing Zhang, Peilin Zhao, and Yatao Bian. Right question is already half the answer: Fully unsupervised llm reasoning incentivization. arXiv preprint arXiv:2504.05812, 2025.

---

> > ### Comment · Reviewer_TNZX · 2025-11-27
> >
> > I thank the authors for their responses. My updated comments are as follows.
> > - It is hard to imagine that there is a typo (sign flip) in the single most important equation which defines the objective of the whole algorithm. BTW, if you check your response, you can find a latex formula does not render...
> > - I am still skeptical about the claim of increasing self-uncertainty improves T2I. In fact, your new experiment "directly optimizing NSC", which trains the model to increase self-uncertainty with gradient, shows exactly the opposite story. That is becasue directly optimizing NSC with gradient **effectively** increases self-uncertainty, which causes the model to collapse. This experiment simply rebuts your own claim. I think the actual situation in your main experiments is that slightly increasing self-uncertainty can improve T2I. However, then the question is what level of self-uncertainty is good, which is a more important question to investigate rather than just claiming increasing self-uncertainty is good.

---

> ### Author Response · Authors · 2025-11-27
>
> (1) I understand your concern. Therefore, we upload our training code, where we implement the self-uncertainty (NSC) in the function ```_get_per_token_self_uncertainty``` in ```IRIS_public/src/t2i-r1/src/open_r1/trainer/grpo_trainer.py```
>
> (2) We agree to your argument. Since we think it is obvious that maximizing the self-uncertainty too much will make the model collapse, i.e., the fully random distribution has the largest self-uncertainty, but will not generate meaningful images. This is why we should use RL to optimize but not directly maximizing. RL maximizes NSC more conservatively: GRPO employs a more conservative strategy by first generating a batch of responses and then aligning with the response with the largest NSC. In our case, each trajectory should first correspond to a reasonable image and only then be optimized, but directly optimizing the NSC would destroy the probabilistic structure of the trajectory and quickly lead to collapse. So to answer your question "what level of self-uncertainty is good", we believe the "largest possible self-uncertainty that the base model can generate" is good.
>
> (3) Regarding your question on random seed. We run the experiments on four different random seeds, and just collected the experiment results. We report the mean and standard deviation in the updated Figure 3 and Table 1. We observe that our experiment results are stable. In particular, training with only intrinsic reward can achieve consistent performance improvement over the base model and comparable to external rewards. We hope this can address your concern.

---

### Official Review · Reviewer_ggZ2 · 2025-11-01

**Soundness:** 3
**Presentation:** 3
**Contribution:** 3
**Rating:** 6
**Confidence:** 4

**Summary:**

This paper introduces IRIS (Intrinsic Reward Image Synthesis), a reinforcement learning framework designed to improve text-to-image generation without relying on external human feedback or labeled data. The key idea is to use intrinsic rewards derived from the model’s own self-uncertainty, showing that maximizing uncertainty—rather than minimizing it—leads to more diverse, detailed, and human-preferred images. The authors demonstrate that autoregressive T2I models with low uncertainty tend to produce oversimplified and uniform results, while optimizing intrinsic uncertainty enhances visual richness and alignment with user intent. Empirical experiments confirm that IRIS effectively improves generation quality.

**Strengths:**

1. This paper provides a interesting observation of the relationship between model confidence and reward in the training process, and the different patterns between text generation and image generation.

2. This paper proposes a reinforcement learning method that improves text-to-image generation without external reward or human annotation. After training with intrinsic reward, the model outperforms the baseline model by a large margin, and achieves comparable performance with external reward-guided training.

3. This paper is motivated by a observation presented in Figure2, and fluently leads to the method design.

**Weaknesses:**

1. The performance using intrinsic reward is slightly lower than using external reward. It is sometimes acceptable to achieve slightly lower performance without the need of external reward, but in some cases people may prefer a higher overall performance.

2. Captions of figure 5 and 6 need to switch. The captions of some other figures also seem unclear. It would be better to summarize the key observations of each figure in the caption.

3. In the experiment, the authors compare the best checkpoints of T2I-R1 and IRIS, which may lead to unfair comparison. It would be better to compare them at the same training steps. Since RL training are often unstable and costly, sometimes the training step is also an important factor that we want to compare. Besides, in figure 3, it seems T2I-R1 consistently outperform IRIS by a large margin. This does not well align with the claimed results in Table1.

**Questions:**

NA

---

> ### Author Response · Authors · 2025-11-21
>
> Thank you for providing your valuable feedback. We have answered your questions in the following and updated the paper accordingly.
>
> 1. **Performance of IRIS** (Weakness 1)
>
> Admittedly, using intrinsic rewards can hardly better performance than using external ones overall. However, analyzing the intrinsic reward can help us better understand the differing behavior of autoregressive generation models in text-to-image and math reasoning tasks. This work highlights the possibility of self-improvement in text-to-image models. Besides, given that training only intrinsic reward can achieve comparable performance with external ones, this phenomenon indicates that the base model itself can reward its own output as effectively as an external reward model. This calls for a better design of external reward models. In addition, the different behavior of autoregressive generation models in text-to-image tasks and math reasoning tasks, i.e., minimizing or maximizing the self-certainty, indicates model's different inclination in its own prediction in different tasks. See **Common Response: Motivation**.
>
> 2. **Typos and figure captions** (Weakness 2)
>
> Thank you for pointing it out. We have revised Figure 5,6's captions in the updated version. We also add summarization the key observations of Figure 2,3,4. The caption of each figure in the ablation studies (Section 4) is the key observation of the corresponding paragraph.
>
> 3. **Experiment results** (Weakness 3)
>
> Regarding to the concern on training cost, in Figure 3, we plotted the evaluation results on three benchmark every 100 steps. In Figure 3, on T2I-Compbench and WISE, our method achieves results comparable to the baseline during all training steps, demonstrating that it does not require additional reward models. Regarding to the concern on performance. Meanwhile, only on Geneval datasets, T2I-R1 outperforms IRIS. On the other two benchemarks, as well as a newly added TIIF-Bench (See Appendix B.7), which are more up-to-date, comprehensive and difficult, IRIS's performance is also comparable with T2I-R1, and achieves better performance in TIIF-Bench given short descriptions.

---

### Official Review · Reviewer_ZMs2 · 2025-11-01

**Soundness:** 3
**Presentation:** 3
**Contribution:** 3
**Rating:** 6
**Confidence:** 3

**Summary:**

This paper introduces IRIS (Intrinsic Reward Image Synthesis), a reinforcement learning framework for text-to-image (T2I) generation that uses negative self-certainty (NSC) as an intrinsic reward. Unlike RLHF or external reward-based approaches, IRIS optimizes autoregressive T2I models without human labels or domain-specific verifiers. The key insight is that reducing self-confidence in image token predictions enhances visual richness and diversity. Experiments on Janus-Pro across GenEval, T2I-CompBench, and WISE benchmarks show IRIS performs competitively or better than models trained with external rewards.

**Strengths:**

[+] Challenges standard RL intuition demonstrates that minimizing self-confidence improves image diversity.

[+] Defines intrinsic rewards via forward KL divergence; mathematically grounded and well-motivated.

[+] Broad evaluation across multiple benchmarks and ablations (text vs. image tokens, KL direction, semantic CoT).

**Weaknesses:**

[-] Reward aggregation details (text/image scaling, normalization) are unclear.

[-] Not includes the comparison with preference- or diffusion-based RL.

[-] There are no human studies to demonstrate that visual preference.

**Questions:**

1. How are text and image token rewards normalized or weighted—are they on comparable scales?
1. Has IRIS been tested on diffusion or masked models beyond autoregressive Janus-Pro?
1. Any user preference studies confirming that lower self-confidence yields subjectively better images?

---

> ### Author Response · Authors · 2025-11-21
>
> Thank you for providing your valuable feedback. We have answered your questions in the following and updated the paper accordingly.
>
> 1. **Reward aggregation details** (Weakness 1 and Question 1)
>
> Please see **Common Response: Reward aggregation details**
>
> 2. **Beyond auto-regressive models**. (Weakness 2 and Question 2)
>
> Please see **Common Response: Beyond auto-regressive models**
>
> 3. **Human preference survey**. (Weakness 3 and Question 3)
>
> Thanks for your suggestion. Please see **Common Response: Human preference survey**.

---

### Author Response · Authors · 2025-11-21
**Common Response**

Thanks for the reviewers for the valuable feedbacks. We address some common questions here.

- **Motivation and contributions**

We updated our contributions in Section 1.

    - We propose IRIS, the first RL-based alignment method for text-to-image generation using only an intrinsic reward}. We minimize the self-certainty of the T2I models.
    - We observe and confirm that the model's self-certainty exhibits task-dependent behaviors. Specifically, higher self-certainty benefits the model in domains requiring objective reasoning (e.g., mathematical and code reasoning), but lower self-certainty benefits the model in subjective generation tasks (e.g., text-to-image generation).
    - Empirically, we show that IRIS can improve the autoregressive T2I models without any external supervision. Post-training by IRIS achieves competitive performance compared to training with external rewards. We call for better design of training algorithm and external reward models that are beyond the intrinsic capabilities of the base T2I model.

- **Typo correction**

Sorry for the confusion. There is a typo in Equation (2): $${\rm SC}(o_t|q,o_{<t}) := +{\rm KL}(U\|\pi_{\theta}(o_{t}|q,o_{<t}))$$ is the correct definition. This definition is aligned with the paper's discussion.


- **Reward aggregation details**
We have discussed in Section 3.2 that
```
For the image tokens, as we discussed in~\cref{fig:Intro}, overly confident models usually generate uniform and plain figures, whereas models with a moderate confidence can generate images with richer and more diverse features. For text tokens, we argue that maximizing NSC encourages the generation of more diverse semantic CoTs, thereby facilitating better exploration during training.
```
Sorry for the confusion for the reviewers. We make it more clear here.
In the autoregressive T2I models, the generated output $o_i$ consists of text and image tokens, denoted by Let $o_{i,\text{text}}$ and $o_{i,\text{img}}$ respectively. We simply concatenate $o_{i,\text{text}}$ and $o_{i,\text{img}}$ to be $o_t$. Therefore, our objective is to maximize the uncertainty (NSC) for both text and image tokens. We have added this discussion in Section 3.2.

- **Additional benchmarks**
We conduct experiments on TIIF-Bench. TIIF-Bench (Text-to-Image Instruction Following Benchmark) is a comprehensive, and difficulty-graded benchmark designed to assess modern T2I models' ability to follow complex textual instructions, addressing limitations like simplistic prompts and coarse evaluation found in prior benchmarks. It has three major catgeries:{Basic Following} ({Attribute}, {Relation}, {Reasoning}), {Advanced Following} ({Attribute+Relation}, {Attribute+Reasoning}, {Relation+Reasoning},{Style},{Text}), and {Real World Following}. For each prompt, this benchmark contains the {short} and {long} versions to more systematically evaluate the model's ability to follow short and long instructions. Please see the entire results in Table B.7. In summary, IRIS achieves better performance given short prompts (IRIS (69.64) vs T2I-R1 (64.18) vs base (61.29)).

- **Beyond auto-regressive models**

Our method is currently limited to autoregressive (AR) text-to-image models. The primary reason is that its foundation in RL requires calculating the next-token probability distribution. Applying RL to other types of text-to-image models, such as flow matching or masked models, is challenging. AR models simplify this because they directly model the per-token log-probabilities. Consequently, the sequence-level log-probability $\left(\log\pi_{\rm AR}(o|q)=\sum_{i=1}^t\log\pi_{\rm AR}(o_t|q,o_{t-1})\right)$ can be easily computed through the chain rule using a single forward pass.

Recently, [2] proposed Flow-GRPO, which first integrates policy gradient RL into flow matching models. However, these models do not operate on discrete tokens like Janus-Pro or Skywork-UniPic. Instead, their update rule involves continuous values, as seen in $x_{t+\delta t}-x_t=\mu(x_t,t) \delta t + \mathcal{N}(0, \sigma_t\sqrt{\delta t})$. This means the policy is an isotropic Gaussian distribution with fixed $\sigma_t$, and its forward KL divergence with respect to the uniform distribution is a non-parameterized constant that cannot be optimized directly. However, there is a growing body of work that focuses on manipulating $\sigma_t$ directly to achieve better exploration during GRPO post-training [4,5]. Although our method approaches the problem from a different perspective, we share the same underlying motivation of enhancing exploration.

Furthermore, masked text-to-image models lack an autoregressive structure and thus do not have the sequential factorization of the sequence log-probability. Therefore, current AR-based RL algorithms, such as GRPO, cannot be directly applied to them. Addressing these models with RL, as seen in works like [2] and [3], is a research direction separate from our current focus.

---

### Author Response · Authors · 2025-11-21
**Common Response (continue)**

- **Human preference survey**

Unluckily we are unable to conduct large scale human preference survey in a university research lab. We conduct a small scale human preference collection. Besides, since the reward models we used, such as HPSv2, are trained on human preferences, we believe can represent the general human preference on images.

For the human evaluation, we randomly sample 100 prompts from the three benchmarks (WISE, T2I-Compbench, GenEval). Due to limited resources, we only evaluated three checkpoints: (1) *Janus-Pro-1B*: base generation model, (2) *T2I-R1* (external reward): Janus-Pro-1B finetuned with four external rewards, and (3) *IRIS* (internal reward): Janus-Pro-1B finetuned with our intrinsic reward. For (2) and (3), we use the best checkpoints selected on WISE. We then ask 9 human evaluators to choose the best image among the three for each prompt and record the number of times each model is upvoted. The following table shows that our method achieves performance comparable to the external-reward-based model.

| Method             | Janus-Pro-1B | T2I-R1 External reward | IRIS (Internal reward) |
|--------------------|-------------:|------------------------:|-----------------------:|
| Upvoting Rate               |         0.13 |                    0.42 |                   0.45 |


[1] Xuandong Zhao, Zhewei Kang, Aosong Feng, Sergey Levine, and Dawn Song. Learning to reason without external rewards. arXiv preprint arXiv:2505.19590, 2025b.
[2] Jie Liu, Gongye Liu, Jiajun Liang, Yangguang Li, Jiaheng Liu, Xintao Wang, Pengfei Wan, Di Zhang, and Wanli Ouyang. Flow-grpo: Training flow matching models via online rl. arXiv preprint arXiv:2505.05470, 2025.
[3] Siyan Zhao, Devaansh Gupta, Qinqing Zheng, and Aditya Grover. d1: Scaling reasoning in diffusion large language models via reinforcement learning. arXiv preprint arXiv:2504.12216, 2025a
[4] Li, J., Cui, Y., Huang, T., Ma, Y., Fan, C., Yang, M., & Zhong, Z. (2025). Mixgrpo: Unlocking flow-based grpo efficiency with mixed ode-sde. arXiv preprint arXiv:2507.21802.
[5] He, X., Fu, S., Zhao, Y., Li, W., Yang, J., Yin, D., ... & Zhang, B. (2025). Tempflow-grpo: When timing matters for grpo in flow models. arXiv preprint arXiv:2508.04324.

---

### Author Response · Authors · 2025-12-03
**Cover Letter of Rebuttal Summarization to Area Chair**

Dear Area Chair,

We sincerely thank your dedication assessing our work in this hard time. With the original responses and updated manuscript, we hope this letter can summarize our improvements and resolutions for all concerns.
### Strengths
We appreciate the reviewers for recognizing:
- **Reviewer ZMs2** (rating 6): "challenges standard RL intuition", "intrinsic rewards via forward KL divergence", "broad evaluation across multiple benchmarks and ablations".
- **Reviewer ggZ2** (rating 6): "different patterns in text vs. image", "gain without external supervision", "method design fluently motivated by empirical observation".
- **Reviewer TNZX** (rating 2): "intrinsic reward without human labels", "lightweight and scalable to new domains", "clear objective for easy reproduction".
- **Reviewer kaX6** (rating 2): "well written and clearly communicated", "strong ablation studies", "novel insight in mode-covering behavior".
### Common Response
- **Contributions**. We summarize our contributions here.
    - **Contribution 1**: We propose IRIS, the first RL-based alignment method for text-to-image generation using only an intrinsic reward}. We minimize the self-certainty of the T2I models.
    - **Contribution 2**: We observe and confirm that the model's self-certainty exhibits task-dependent behaviors. Specifically, higher self-certainty benefits the model in domains requiring objective reasoning (e.g., mathematical and code reasoning), but lower self-certainty benefits the model in subjective generation tasks (e.g., text-to-image generation).
    - **Contribution 3**: Empirically, we show that IRIS can improve the autoregressive T2I models without any external supervision. Post-training by IRIS achieves competitive performance compared to training with external rewards. We call for better design of training algorithm and external reward models that are beyond the intrinsic capabilities of the base T2I model.

- **Reward aggregation details** (ZMs2 W1Q1, TNZX Q2). We clarify that we concatenate text and image tokens directly and maximize the uncertainty (NSC) for both text and image tokens.
- **Additional benchmarks** (kaX6 W2Q1). We add TIIF benchmark, and show that IRIS outperforms external rewards on short instructions.
- **Beyond auto-regressive models** (ZMs2 W2Q2, kaX6 W2Q3). Our method is currently limited to autoregressive (AR) text-to-image models, because the RL algorithm (GRPO) and the intrisic reward (NSC) that we used require the sequential factorization of the sequence probability. Adapting RL algorithm and the intrisic reward to other text-to-image model structures, such as masked or diffusion models, is a nontrivial task and independent of our primary contribution.
- **Human preference survey** (ZMs2 W3Q3, TNZX W6, kaX6 W1). We perform a human study on 100 samples. Results show that IRIS is preferred by human.

### Response to Reviewer ZMs2
We address the reviewr ZMs2's comments in the common response.
### Response to Reviewer ggZ2
- **Performance of IRIS & Experiment results** (W1W3). We show that although on GenEval, IRIS underperforms external rewards, on all more up-to-date and difficult benchmarks, including T2I-CompBench, WISE, and TIIF, IRIS is comparable and could be better than external rewards, reaffirming Contribution 3.
### Response to Reviewer TNZX
- **Definition** (W1W2W3Q1,Q1(2nd response)). There is a typo in Equation (2): ${\rm SC}(o_t|q,o_{<t}) := +{\rm KL}(U\|\pi_{\theta}(o_{t}|q,o_{<t}))$ is the correct definition of SC. This definition is aligned with the paper's discussion, and we add our implementation code to the suppplementary to support that.
- **Novelty** (W4). We state in Contribution 1&2 that our work is novel and different from exisitng RLIF works.
- **More steps** (Q3). We show that IRIS's performance drop after longer training. However, it is a common shortcoming of the RLIF and not limited to our method.
- **Directly optimizing NSC** (Q4, Q2 (2nd response)). We show that using RL to maximize NSC is better than directly setting NSC as an objective to maximize. There might be misunderstanding but the result does not refute the claim in Contribution 2. Higher uncertainty benefits exploration does not means fully random (highest uncertainty) is good. We explain this difference by: GRPO employs a more conservative strategy by first generating a batch of responses and then aligning with the best-performing output. In contrast, directly optimizing NSC can lead to more aggressive updates that may distort the probability structure of the trajectory.  Regarding "what level of self-uncertainty is good", we believe the "largest possible self-uncertainty that the base model can generate" is good.

---

> ### Author Response · Authors · 2025-12-03
> **Cover Letter of Rebuttal Summarization to Area Chair (cont'd)**
>
> ### Response to Reviewer kaX6
> - **Mechanism** (W3 & Q2). We explain the phenomenon in the following: increased self-certainty typically results in more uniform and less visually diverse images, while lower self-certainty tends to generate images with richer visual features that are more preferred by humans
> - **Increase in entropy on text tasks** (W4). We reformulate the Contribution 2 according to the additional evidence the reviewr provided. The evidence is consistent with our claim that exploratory tasks favors more uncertain models.
>
> ### Summary
> In this rebuttal, we have remarkably enriched the paper with 4 additional experiments (human eval, error bar, direct entropy maximization, continuous training experiment), and many discussions inspired be the reviewers. We believe IRIS stands as a valuable and insightful contribution to the text-to-image generation community.
>
> Best regards,
>
> The Authors of Submission 23590

---

### Meta-Review · Area_Chair_pkK4 · 2026-01-01

**Summary:**

The paper proposes IRIS (Intrinsic Reward Image Synthesis), a framework for fine-tuning autoregressive text-to-image (T2I) models using intrinsic rewards. The core contribution is the introduction of "Negative Self-Certainty" (NSC)—effectively incentivizing higher entropy/uncertainty in the model's output distribution—optimized via Group Relative Policy Optimization (GRPO). The authors posit that unlike reasoning tasks where certainty is preferred, image generation benefits from uncertainty to promote diversity and visual richness.

While the reviewers acknowledged the interesting counter-intuitive premise (maximizing uncertainty) and the extensive benchmarking, the submission is recommended for rejection. The decision is primarily driven by significant concerns regarding the theoretical soundness of the objective (directly optimizing the proposed reward leads to model collapse), the marginal performance gains compared to external reward baselines, and unresolved debates regarding the definitions and mechanisms driving the reported improvements.

**Reviewer Concerns:**

Addressed Concerns:
- Human Evaluation: The authors responded to requests for human preference studies (Reviewers ZMs2, kaX6) by providing a small-scale study (N=100, 9 evaluators) suggesting IRIS is competitive.
- Clarification of Definitions: The authors corrected a typo in Equation 2 regarding the definition of Self-Certainty (SC) and Negative Self-Certainty (NSC) to address Reviewer TNZX’s confusion regarding the sign of the KL divergence.
- Additional Benchmarking: The authors added the TIIF-Bench results to demonstrate performance on instruction following, addressing concerns about limited evaluation scope.

Outstanding Concerns:
- Theoretical Validity and Optimization: Reviewer TNZX raised a critical point that the authors' ablation study—where directly optimizing NSC using gradient descent caused model collapse—undermines the central claim. If increasing self-uncertainty is the goal, direct optimization should work. The fact that it fails suggests the method relies heavily on the implicit regularization of the specific RL algorithm (GRPO) rather than the intrinsic quality of the reward signal itself.
- Marginal Utility: Reviewers ggZ2 and kaX6 noted that IRIS generally performs comparably to or slightly worse than methods using external rewards. The motivation to use a complex RL scheme with intrinsic rewards is weak if it does not offer a clear performance advantage or robustness benefit over established external reward models.
- Mechanism Attribution: There remains skepticism (Reviewer TNZX) about whether the gains are truly due to the NSC reward or the "Semantic CoT" (Chain of Thought) reasoning step used in the pipeline. The disentanglement of these factors remains unconvincing.
- Generalization: Reviewer kaX6 maintained that the contribution is limited by its restriction to autoregressive models (Janus-Pro), limiting its impact in a field currently dominated by diffusion and flow-matching models.
- Conflicting Literature: Reviewer kaX6 pointed out conflicts with recent literature (e.g., Tongyi DeepResearch) suggesting entropy increases with reward in text tasks as well, complicating the paper's narrative that T2I requires a fundamentally different uncertainty regime than text generation.

**Reviewer Scores:**

- Reviewer ZMs2 (Score: 6 → 5): This reviewer was initially positive about the "challenge to standard RL intuition." However, seeing the discourse regarding the "direct optimization" failure and the limitations in the human study size would likely lower their confidence in the robustness of the method.

- Reviewer ggZ2 (Score: 6 → 5): This reviewer highlighted the issue that IRIS underperforms external rewards. The rebuttal confirmed that IRIS is at best "comparable," which likely reinforces the reviewer's concern about the practical utility of the method, leading to a slight downgrade.

- Reviewer TNZX (Score: 2 → 2): This reviewer remained strongly opposed. The rebuttal confirmed their suspicion that maximizing the proposed metric directly leads to collapse. They explicitly stated in the discussion that they remain skeptical of the core claim. Their score would remain unchanged.

- Reviewer kaX6 (Score: 2 → 2): This reviewer explicitly maintained their rejection stance after the rebuttal, citing insufficient contribution for a conference paper and a lack of convincing evidence that this approach solves a failure mode of existing reward models. Their score would remain unchanged.

---

### Decision · Program_Chairs · 2026-01-26

Reject